# PESD-TSF: A Period-Aware and Explicit Structured Decomposition Framework for Long-Term Time Series Forecasting

Hua Wang [1]   Xianhao Jiao [1]   Fan Zhang [2]

## Abstract

Deep forecasting models often suffer from attenuated periodic perception and entangled trend–noise representations as network depth increases. Moreover, the widely adopted channel-independent paradigm, while improving training stability, disrupts intrinsic dynamic coordination among variables, hindering the modeling of cross-variable consistency in multivariate time series. To address these issues, we propose PESD-TSF, a physics-inspired structured decomposition framework for long-term time series forecasting that jointly emphasizes interpretability and predictive accuracy. PESD-TSF introduces three key designs. First, a Multiplicative Periodic Gating mechanism incorporates continuous-time priors to dynamically modulate signal amplitudes, preserving periodic structures across deep layers. Second, a multi-scale structured encoder integrates detrended attention with hierarchical sampling to explicitly decouple long-term trends from high-frequency variations while retaining fine-grained temporal semantics. Third, to recover disrupted inter-variable dependencies, we propose Cross-Scale Collaborative Attention (CSCA) together with an RLC regularization scheme, which reconstructs global inter-variable topology in deep feature spaces and enforces physically consistent collaboration through orthogonality and consistency constraints. Extensive experiments on benchmark datasets from multiple domains demonstrate that PESD-TSF consistently achieves state-of-the-art performance, with particularly strong gains on multivariate forecasting tasks involving complex inter-variable coupling, highlighting its supe-

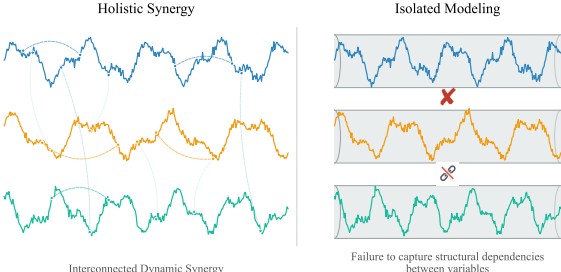

*Figure 1.* Capturing vs. Losing Dependencies. Left: Inherent dynamic synergy in multivariate series. Right: Channel-independent methods sever these crucial connections, hindering consistent feature learning.

rior structural modeling capability and generalization.Code is available at this repository: `https://github.com/bxljsf/PESD-TSF`

## 1. Introduction

Multivariate long-term time series forecasting (LSTF) (Vaswani et al., 2017) is critical for domains like energy and traffic management (Qian et al., 2019). While deep learning models have surpassed traditional statistical approaches in accuracy, they often operate as "black boxes" lacking physical inductive bias (Zhang et al., 2026b; Wang et al., 2026b). Consequently, these models struggle to distinguish robust physical regularities (e.g., periodicity, inter-variable dependency) from stochastic noise, leading to brittle predictions under distribution shifts (Wang et al., 2025). Specifically, existing architectures face three limitations: (1) Signal Dilution: Static positional encodings fail to preserve periodic patterns (e.g., calendar effects) (Luo & Wang, 2024b) within deep feature hierarchies; (2) Noise Entanglement: Attention mechanisms are often distracted by abrupt perturbations, hindering the separation of long-term trends from noise; and (3) Loss of Correlation: As shown in Figure 1, the prevalent channel-independent strategy artificially severs intrinsic cross-variable dependencies (Xue et al., 2023; Zhang et al., 2026a), failing to model dynamic system coordination.

[1]School of Computer and Artificial Intelligence, Ludong University, Yantai, Shandong 264025, China. [2]School of Computer Science and Technology, Shandong Technology and Business University, Yantai, Shandong 264005, China. Correspondence to: Fan Zhang <zhangfan@sdtbu.edu.cn>.

*Proceedings of the 43$^{rd}$ International Conference on Machine Learning*, Seoul, South Korea. PMLR 306, 2026. Copyright 2026 by the author(s).

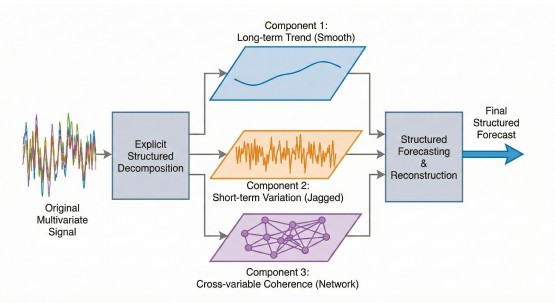

*Figure 2.* Schematic of PESD-TSF. The framework explicitly decomposes signals into three physical components: (a) long-term trend, (b) short-term variations, and (c) cross-variable coherence. This structured decomposition unifies high forecasting accuracy with physical interpretability.

To address these challenges, we propose PESD-TSF, a deep decomposition framework guided by physical inductive bias. As illustrated in Figure 2, PESD-TSF explicitly decomposes temporal signals into three dimensions: long-term trends, short-term perturbations, and cross-variable collaborations. First, to mitigate periodic attenuation, we design a Multiplicative Periodic Gating mechanism that modulates signal amplitudes using continuous-time priors. Second, we construct a multi-scale structured encoder employing detrended attention and hierarchical sampling to disentangle trends from local variations. Finally, to recover inter-variable dependencies, we introduce Cross-Scale Collaborative Attention (CSCA) with RLC regularization. This module enforces orthogonality and consistency constraints, guiding the model to learn structurally consistent topological dependencies. Extensive experiments demonstrate that PESD-TSF achieves state-of-the-art performance, effectively unifying high forecasting accuracy with physical interpretability.

In summary, the main contributions of this paper are as follows:

- We propose **PESD-TSF**, a framework that explicitly decomposes temporal dynamics into three physical dimensions—trend, perturbation, and collaboration—to effectively address long-range dependency attenuation and dynamic variable coupling.

- We design a **Multi-scale Structured Encoder** integrated with **Periodic Gating**. This architecture injects continuous temporal priors to mitigate signal dilution and employs hierarchical sampling to disentangle multi-granularity features.

- We introduce a **Regularized Latent Component (RLC)** module. By imposing orthogonality constraints, it enforces physically interpretable, disentangled representations and reconstructs cross-variable topological consistency.

- Extensive experiments across diverse benchmarks demonstrate that PESD-TSF achieves **state-of-the-art performance**, validating the superiority of explicit structural decomposition over purely stacking deep architectures.

## 2. Methodology

To address long-term dependency attenuation and neglected cross-variable correlations in time series forecasting, we propose PESD-TSF, a framework grounded in physical inductive bias. Departing from generic "black-box" paradigms, PESD-TSF employs a structured pipeline—period enhancement, multi-scale decoupling, collaborative reconstruction, and regularization—to explicitly capture underlying sequence dynamics.

### 2.1. Problem Definition and Overall Architecture

We map historical observations $X \in \mathbb{R}^{B \times L \times C}$ to future states $Y \in \mathbb{R}^{B \times O \times C}$. As shown in Figure 3, PESD-TSF employs a hierarchical architecture starting with Instance Normalization and Periodic Gating to modulate amplitudes via temporal priors. Subsequently, a three-stage encoder disentangles trends, local variations, and correlations, followed by a linear decoder supervised by an RLC module for structural constraints.

### 2.2. Period-Aware Gating and Multi-Scale Embedding

Standard additive positional encoding suffers from Signal Dilution and lacks physical semantics. We propose Multiplicative Periodic Gating to model temporal attributes as continuous priors. Features $\mathcal{M} \in \mathbb{R}^{B \times L \times N_{\text{freq}}}$ are mapped to embedding subspace $\mathbf{W}_{\text{emb}}^{(k)} \in \mathbb{R}^{V_k \times D_{\text{emb}}}$:

$$E_{b,t}^{(k)} = \mathbf{W}_{\text{emb}}^{(k)}[m_{b,t,k}] \in \mathbb{R}^{D_{\text{emb}}}. \tag{1}$$

Embeddings are concatenated into $E_{\text{cat}}$ and projected to context $E_{\text{time}}$ via $W_{\text{fuse}}$:

$$E_{\text{cat}} = \text{Concat}(E^{(1)}, E^{(2)}, \dots, E^{(N_{\text{freq}})}), \tag{2}$$

$$E_{\text{time}} = E_{\text{cat}} W_{\text{fuse}} \in \mathbb{R}^{B \times L \times D}, \tag{3}$$

where $E_{\text{cat}} \in \mathbb{R}^{B \times L \times (N_{\text{freq}} \cdot D_{\text{emb}})}$ and $D_{\text{emb}}$ is the embedding dimension. $W_{\text{fuse}}$ bridges multi-scale dependencies. A gating network then generates $G_{\text{period}}$:

$$G_{\text{period}} = \sigma(E_{\text{time}} W_{\text{gate}} + b_{\text{gate}}) \in (0,1)^{B \times L \times 1}, \tag{4}$$

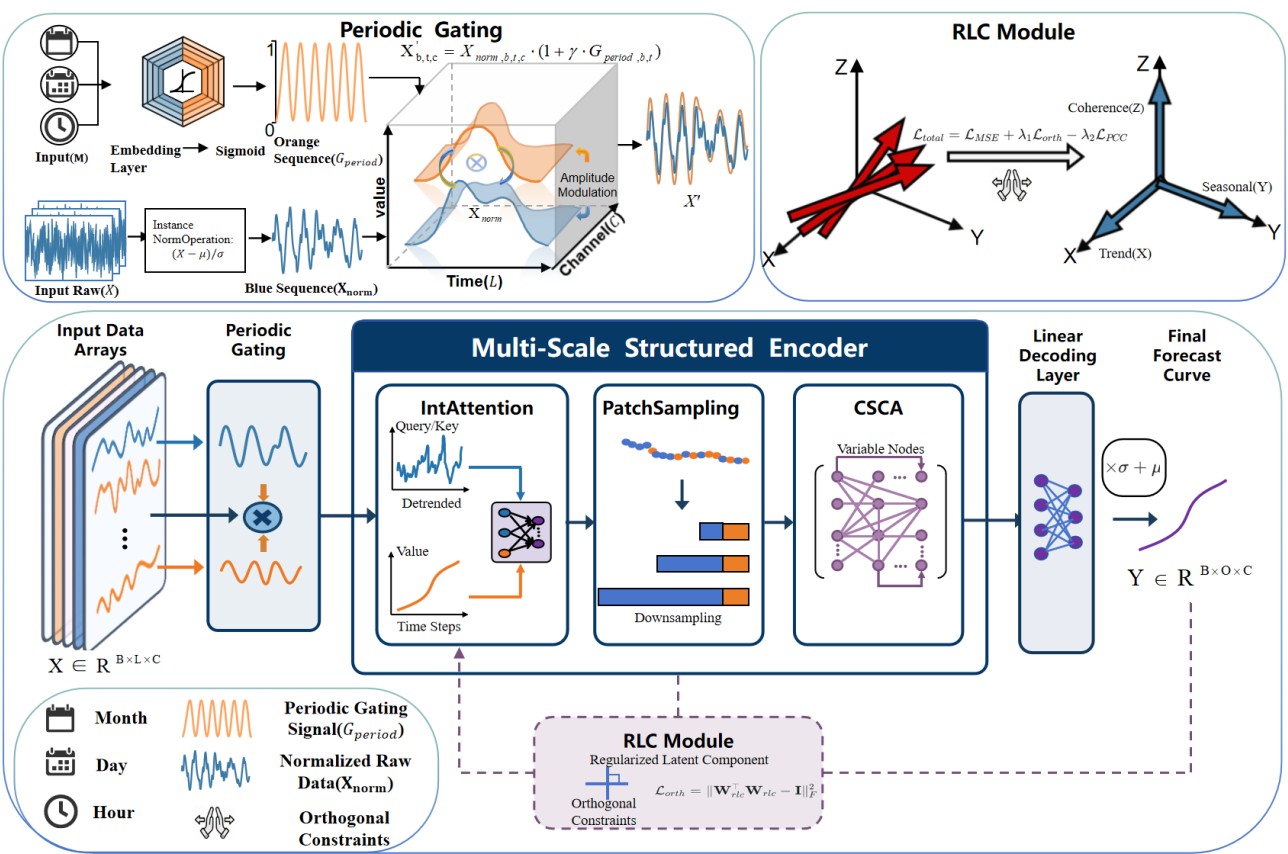

*Figure 3.* The PESD-TSF framework comprises three core components: (1) **Periodic Gating**: fusing temporal features through amplitude modulation to capture periodic patterns; (2) **Multi-scale Structured Encoder**: integrating IntAttention, PatchSampling, and CSCA for multi-resolution feature processing and context aggregation; (3) **RLC module**: utilizing orthogonal constraints ($\mathcal{L}_{\text{orth}}$) to decouple the representations into trend, seasonal, and consistent components. Finally, a linear layer outputs the prediction result $Y$.

where $W_{\text{gate}}, b_{\text{gate}}$ are projection parameters and $\sigma$ is Sigmoid. This signal modulates the normalized input $X_{\text{norm}}$ to yield $X'$:

$$X'_{b,t,c} = X_{\text{norm},b,t,c} \cdot (1 + \gamma \cdot G_{\text{period},b,t}), \qquad (5)$$

where $\gamma$ controls enhancement intensity. Finally, the $n$-th patch $p_{b,n,c} \in \mathbb{R}^P$ of $X'$ is projected to latent space $Z^{(0)}$:

$$Z^{(0)}_{b,n,c} = p_{b,n,c}W_{\text{emb}} \in \mathbb{R}^D, \qquad (6)$$

where $W_{\text{emb}}$ is the projection matrix. Using zero-padding and stride $S$, the total patch count $N$ is:

$$N = \lfloor \frac{L - P}{S} \rfloor + 1. \qquad (7)$$

This yields $Z^{(0)} \in \mathbb{R}^{B \times N \times C \times D}$, fusing local features with periodic priors for the encoder.

### 2.3. Multi-scale Structured Decomposition Encoder

The encoder adopts a cascaded three-stage design with physical inductive biases: IntAttention captures robust temporal

patterns, PatchSampling aggregates multi-scale features, and Cross-Scale Cooperative Attention (CSCA) models inter-variable dependencies. Together, these stages form a coherent decomposition–reconstruction pipeline.

**Stage 1: Integrated Attention (IntAttention) — Denoised Trend Extraction**

Stage 1 receives embedding $\mathbf{Z}^{(0)} \in \mathbb{R}^{B \times N \times C \times D}$ and aims to extract stable trends via Smoothing Convolution. Using a uniform kernel $K_w = \frac{1}{w}1_w$ (with kernel size $w = 3$ and same-padding to maintain sequence length $N$), we calculate the local trend $Z^{(0)}_{\text{trend}}$:

$$Z^{(0)}_{\text{trend}} = Z^{(0)} \circledast K_w \in \mathbb{R}^{B \times N \times C \times D}, \qquad (8)$$

where $\circledast$ denotes temporal convolution. We subtract this to obtain the detrended component $Z^{(0)}_{\text{det}}$ containing high-frequency fluctuations:

$$Z^{(0)}_{\text{det}} = Z^{(0)} - Z^{(0)}_{\text{trend}}. \qquad (9)$$

We adopt an Asymmetric Design. To strictly preserve

channel independence, we reshape the input tensor to $(B \cdot C) \times N \times D$ prior to the attention operation. We then use the detrended component $Z_{det}^{(0)}$ for Queries and Keys to emphasize structural similarity, while Values derive from the complete $Z^{(0)}$. The output $H_{attn}$ is defined as:

$$Q = Z_{\text{det}}^{(0)} W_Q, \quad K = Z_{\text{det}}^{(0)} W_K, \quad V = Z^{(0)} W_V, \quad (10)$$

where $H = 8$ is the number of attention heads. The output is defined as:

$$H_{\text{attn}} = \text{Softmax}\left(\frac{QK^\top}{\sqrt{D/H}}\right) V. \quad (11)$$

Finally, we apply Residual Connections and Layer Normalization to obtain $Z^{(1)}$:

$$Z^{(1)} = \text{LayerNorm}(Z^{(0)} + H_{\text{attn}}) \in \mathbb{R}^{B \times N \times C \times D}. \quad (12)$$

This preserves original information while injecting denoised contextual features.

**Stage 2: PatchSampling — Multi-granularity Feature Aggregation**

To capture long-period dependencies, PatchSampling compresses $Z^{(1)}$ via a dual-branch structure ($Conv1d_{time}$ and $MaxPool_{time}$ with stride 2, kernel 3), mapping concatenated outputs to dimension $D$ without information loss:

$$Z_{\text{agg}} = \mathbf{W}_{\text{agg}}\big(\text{Concat}_{\text{dim}=D}\big[$$
$$\text{Conv1d}_{\text{time}}(Z^{(1)}), \quad (13)$$
$$\text{MaxPool}_{\text{time}}(Z^{(1)})\big]\big),$$

where $\text{Conv1d}_{\text{time}}$ denotes 1D convolution and $\text{Concat}_{\text{dim}=D}$ denotes feature concatenation. This reduces the temporal dimension to $N' = \lfloor N/2 \rfloor$, forming coarse-grained features:

$$Z^{(2)} \in \mathbb{R}^{B \times N' \times C \times D}. \quad (14)$$

This provides multi-scale context for the subsequent CSCA stage.

**Stage 3: Cross-Scale Cooperative Attention (CSCA) - Global Synergy Reconstruction**

Stage 3 inputs $Z^{(2)}$ to extract Global Variable Synergy. Departing from computationally prohibitive dynamic graph modeling, we focus on stable, system-wide correlations. Specifically, we perform Global Average Pooling (GAP) along the time dimension to generate the compact representation $H_c$

$$\mathbf{H}_c = \frac{1}{N'} \sum_{t=1}^{N'} Z_{:,t,:,:}^{(2)} \in \mathbb{R}^{B \times C \times D}. \quad (15)$$

This operation aggregates temporal contexts, allowing the subsequent attention mechanism to discover intrinsic, time-invariant dependencies among variables.

We construct self-attention over the variable dimension $C$ on $\mathbf{H}_c$ to capture implicit dependencies, generating cooperative context $\mathbf{C}_{\text{context}}$:

$$\mathbf{C}_{\text{context}} = \text{Softmax}\left(\frac{(\mathbf{H}_c \mathbf{W}_q)(\mathbf{H}_c \mathbf{W}_k)^\top}{\sqrt{D}}\right)$$
$$\times (\mathbf{H}_c \mathbf{W}_v) \in \mathbb{R}^{B \times C \times D}, \quad (16)$$

where $\mathbf{W}_q, \mathbf{W}_k, \mathbf{W}_v$ are projection matrices. Finally, we inject this global structure back into the local stream via a Structured Conditioning mechanism to obtain $Z_{\text{final}}$:

$$Z_{\text{final},b,t,c} = Z_{b,t,c}^{(2)} + C_{\text{context},b,c}. \quad (17)$$

This forces local dynamics to satisfy system-wide topological constraints while avoiding expensive dynamic graph calculations.

## 2.4. Final Prediction and Regularization Constraints

To mitigate autoregressive error accumulation, we adopt a 'Mixed-Encoding, Independent-Decoding' strategy, flattening the temporal and embedding dimensions of $Z_{\text{final}}$ to construct the regression input:

$$\tilde{Z} = \text{Flatten}(Z_{\text{final}}) \in \mathbb{R}^{B \times C \times (N \cdot D)}, \quad (18)$$

where $B$, $N$, $C$, and $D$ denote the batch size, temporal length, number of variables, and embedding dimension, respectively. A channel-shared linear projection $W_{\text{head}} \in \mathbb{R}^{(N \cdot D) \times O}$ is then applied to obtain the prediction:

$$H_{\text{pred}} = \tilde{Z} W_{\text{head}} \in \mathbb{R}^{B \times C \times O}. \quad (19)$$

The output is transposed to $\mathbb{R}^{B \times O \times C}$ to enable independent variable extrapolation. We then apply exact inverse normalization using preserved statistics $\mu_X$ and $\sigma_X$ to restore the physical scale:

$$\hat{Y} = H_{\text{pred}} \odot \sigma_X + \mu_X, \quad (20)$$

where $\odot$ denotes the Hadamard product, and $\mu_X, \sigma_X$ are replicated along the forecasting horizon $O$ for dimensional alignment.

**Regularized Latent Component (RLC) as Auxiliary Statistical Alignment** To further enhance robustness and prevent overfitting, we introduce the Regularized Latent Component (RLC) module as an auxiliary regularization constraint operating on the *statistical readout* of latent representations. Unlike post-hoc analysis, the RLC module is fully integrated into the training graph and backpropagates gradients into the encoder.

Concretely, given the encoder output $Z_{\text{final}} \in \mathbb{R}^{B \times N \times C \times D}$, we compute channel-wise statistical descriptors by aggregating over the temporal and embedding dimensions:

$$\mu(Z_{\text{final}}) = \frac{1}{ND} \sum_{n=1}^{N} \sum_{d=1}^{D} Z_{\text{final}}[:, n, :, d], \quad (21)$$

$$\sigma(Z_{\text{final}}) = \sqrt{\frac{1}{ND}\sum_{n,d}\left(Z_{\text{final}}[:,n,:,d] - \mu(Z_{\text{final}})\right)^2}.$$

(22)

The resulting statistical feature vector is defined as:

$$F_{\text{stat}} = \text{Concat}(\mu(Z_{\text{final}}), \sigma(Z_{\text{final}})) \in \mathbb{R}^{B \times 2C}. \quad (23)$$

Similarly, we extract pooled future statistics from the ground-truth targets:

$$Y_{\text{pool}} = \text{Concat}(\text{Mean}(Y_{\text{gt}}), \text{Std}(Y_{\text{gt}})) \in \mathbb{R}^{B \times 2C}, \quad (24)$$

where statistics are computed over the forecasting horizon.

The RLC module projects $F_{\text{stat}}$ into a compact latent factor space:

$$Z_{\text{rlc}} = F_{\text{stat}}W_{\text{rlc}}, \quad W_{\text{rlc}} \in \mathbb{R}^{2C \times K}. \quad (25)$$

To encourage the extracted factors to capture distinct and complementary statistical patterns, we impose a column orthogonality constraint on the projection matrix:

$$\mathcal{L}_{\text{orth}} = \|W_{\text{rlc}}^\top W_{\text{rlc}} - I\|_F^2, \quad (26)$$

where $K \leq 2C$ is enforced to avoid rank deficiency and information redundancy. This constraint ensures that the linear projection does not introduce additional correlations or amplify perturbations within the statistical latent space (see Appendix B for detailed analysis).

To further align latent statistical factors with future targets, we maximize the Pearson Correlation Coefficient (PCC) between $Z_{\text{rlc}}$ and $Y_{\text{pool}}$ across the batch dimension:

$$\text{PCC}(\mathbf{u}, \mathbf{v}) = \frac{\sum_{i=1}^B (u_i - \bar{u})(v_i - \bar{v})}{\sqrt{\sum_{i=1}^B (u_i - \bar{u})^2}\sqrt{\sum_{i=1}^B (v_i - \bar{v})^2}}, \quad (27)$$

where $\bar{u}$ and $\bar{v}$ denote batch-wise means. Computing PCC across the batch enforces distributional consistency, ensuring that relative statistical ordering is preserved while discouraging sample-wise memorization.

Finally, the overall training objective is formulated as:

$$\mathcal{L}_{\text{total}} = \mathcal{L}_{\text{MSE}}(\hat{Y}, Y_{\text{gt}}) + \lambda_1 \mathcal{L}_{\text{orth}} - \lambda_2 \sum_{k=1}^K \text{PCC}(Z_{\text{rlc}}^{(k)}, Y_{\text{pool}}^{(k)}), \quad (28)$$

where $\lambda_1$ and $\lambda_2$ balance prediction accuracy, statistical orthogonality, and consistency constraints.

# 3. Experiment

To evaluate PESD-TSF, we conduct experiments on three aspects: (1) long-term forecasting, emphasizing global trends and periodic patterns; (2) short-term forecasting, focusing on high-frequency local dynamics; and (3) ablation studies, assessing the contributions of period-aware gating, cross-scale cooperative attention (CSCA), RLC regularization, and hierarchical decoupling. Detailed settings are provided in Appendix C.

## 3.1. Long-term Forecasting

To validate generalization, we categorize experiments into three groups: (1) **Classic Benchmarks** (Table 1), covering seven standard datasets like ETT, Electricity, and Weather; (2) **Diverse Domain-Specific Datasets** (Table 2), extending to Solar, Air Quality, and Medical domains (Luo & Wang, 2024a); and (3) **High-Dimensional Benchmarks** (Appendix Table 10), comprising Meter, Atec, and Mobility for massive multivariate series. Following standard protocols ($L = 720$), we evaluate using MSE and MAE (details in Appendix D.1). PESD-TSF achieves Top-2 accuracy across most datasets, verifying its ability to capture intrinsic dependencies regardless of domain variations. Compared to DLinear (Zeng et al., 2023), it reduces MSE and MAE by 14.96% and 14.87% respectively. Against lightweight SOTA models (FITS (Xu et al., 2023), SparseTSF (Lin et al., 2025)), it achieves 10.23% ∼ 11.59% error reduction. Notably, PESD-TSF ranks **1st in 35 out of 38** metrics, underscoring superior robustness despite its compact design. Furthermore, Appendix Table 10 confirms superior performance on high-dimensional datasets against specialized baselines. For full results, refer to Appendix E.

## 3.2. Short-term Forecasting

We conducted short-term forecasting experiments on PeMS datasets (Input 96, Output 12) (Wang et al., 2023) and evaluated performance using MAE, MAPE, and RMSE (details in Appendix D.2) (Zhang & Yan, 2023). As shown in Table 3, leveraging its effective modeling of complex spatiotemporal dependencies (Liu et al., 2022b), PESD-TSF demonstrates a distinct performance advantage over channel-independent approaches like PatchTST (Nie et al., 2023) and DLinear (Zeng et al., 2023). Specifically, PESD-TSF achieved state-of-the-art results across all metrics on PeMS03, 07, and 08 (Liu et al., 2022a), comprehensively outperforming TimeMixer (Wang et al., 2024). Even on PeMS04, it secured a significant lead in MAPE (11.62 vs. 12.53), highlighting its dominance in multivariate spatiotemporal modeling tasks (Qiu et al., 2025b).

## 3.3. Ablation Studies

We conducted ablation studies (Table 4) to verify the contribution of each component. **For comprehensive results, please refer to Table 11.** Key observations indicate:

**w/o Period**: Removing period-aware gating significantly increased errors on datasets with calendar effects (e.g., Traffic, ETTh1), confirming that explicit temporal priors prevent signal dilution.

**w/o CSCA**: The removal of this module degraded multivariate forecasting performance (especially on PeMS), highlighting the necessity of modeling dynamic coupling to

*Table 1.* Long-term sequence prediction performance of PESD-TSF. The best and second-best results are highlighted in bold and underlined, respectively.

| | | OURS | | 2025($l_{\text{WEIGHT}}$) | | 2024($l_{\text{WEIGHT}}$) | | 2025 | | 2023-2024 | | | | | | | |
| | | PESD-TSF | | TIMEBASE | | SPARSETSF | | TIMEBRIDGE | | ITRANS* | | DEFORM** | | TIMEMIXER | | PATCHTST | |
| MODELS DATASET | L | MSE | MAE | MSE | MAE | MSE | MAE | MSE | MAE | MSE | MAE | MSE | MAE | MSE | MAE | MSE | MAE |
|---|---|---|---|---|---|---|---|---|---|---|---|---|---|---|---|---|---|
| ETTM1 | AVG | **0.343** | **0.370** | 0.357 | 0.381 | 0.362 | 0.384 | 0.344 | 0.379 | 0.362 | 0.391 | 0.348 | 0.383 | 0.355 | 0.380 | 0.353 | 0.382 |
| ETTM2 | AVG | **0.244** | **0.304** | 0.251 | 0.314 | 0.253 | 0.317 | 0.246 | 0.310 | 0.269 | 0.329 | 0.257 | 0.319 | 0.257 | 0.318 | 0.256 | 0.317 |
| ETTH1 | AVG | **0.394** | 0.421 | 0.396 | **0.415** | 0.407 | 0.419 | 0.397 | 0.424 | 0.439 | 0.448 | 0.404 | 0.423 | 0.427 | 0.441 | 0.413 | 0.434 |
| ETTH2 | AVG | 0.339 | 0.378 | 0.347 | 0.398 | 0.344 | 0.387 | 0.341 | 0.382 | 0.374 | 0.406 | 0.328 | **0.377** | 0.349 | 0.397 | **0.324** | 0.381 |
| WTH | AVG | **0.217** | **0.248** | 0.219 | 0.263 | 0.244 | 0.286 | 0.219 | 0.249 | 0.233 | 0.271 | 0.222 | 0.262 | 0.226 | 0.264 | 0.226 | 0.264 |
| TRAFFIC | AVG | **0.354** | **0.242** | 0.418 | 0.279 | 0.414 | 0.280 | 0.360 | 0.255 | 0.397 | 0.282 | 0.391 | 0.278 | 0.409 | 0.279 | 0.391 | 0.264 |
| ECL | AVG | 0.150 | **0.244** | 0.167 | 0.258 | 0.168 | 0.264 | **0.149** | 0.245 | 0.164 | 0.261 | 0.161 | 0.261 | 0.185 | 0.284 | 0.159 | 0.253 |

\* INDICATES FORMER; \*\* INDICATES ABLETST

*Table 2.* Long-term sequence prediction performance of PESD-TSF. The best and second-best results are highlighted in bold and underlined, respectively.

| | | OURS | | BASELINES | | | | | | | | | | | | | |
| | | PESD-TSF | | TIMEBASE | | SPARSETSF | | TIMEMIXER | | FITS | | ITRANS* | | DLINEAR | | PATCHTST | |
| MODELS DATASET | L | MSE | MAE | MSE | MAE | MSE | MAE | MSE | MAE | MSE | MAE | MSE | MAE | MSE | MAE | MSE | MAE |
|---|---|---|---|---|---|---|---|---|---|---|---|---|---|---|---|---|---|
| AQSHUNYI | AVG | **0.665** | **0.488** | 0.675 | 0.506 | 0.760 | 0.546 | 0.736 | 0.536 | 0.763 | 0.548 | 0.723 | 0.520 | 0.757 | 0.572 | 0.691 | 0.503 |
| AQWAN | AVG | **0.769** | **0.480** | 0.779 | 0.499 | 0.826 | 0.526 | 0.822 | 0.521 | 0.813 | 0.519 | 0.843 | 0.538 | 0.883 | 0.560 | 0.810 | 0.513 |
| CZELAN | AVG | **0.209** | **0.246** | 0.225 | 0.262 | 0.241 | 0.292 | 0.234 | 0.282 | 0.250 | 0.304 | 0.245 | 0.297 | 0.242 | 0.290 | 0.229 | 0.276 |
| ZAFNOO | AVG | 0.502 | **0.435** | **0.495** | 0.445 | 0.531 | 0.491 | 0.505 | 0.464 | 0.521 | 0.480 | 0.540 | 0.500 | 0.540 | 0.495 | 0.510 | 0.469 |
| METR-LA | AVG | **1.181** | **0.657** | 1.254 | 0.754 | 1.302 | 0.764 | 1.271 | 0.741 | 1.296 | 0.760 | 1.253 | 0.734 | 1.301 | 0.759 | 1.210 | 0.706 |
| PM2.5 | AVG | **0.387** | **0.407** | 0.428 | 0.437 | 0.467 | 0.477 | 0.465 | 0.474 | 0.460 | 0.469 | 0.480 | 0.492 | 0.449 | 0.457 | 0.459 | 0.467 |
| SOLAR | AVG | **0.182** | **0.218** | 0.216 | 0.254 | 0.216 | 0.264 | 0.244 | 0.296 | 0.229 | 0.279 | 0.233 | 0.285 | 0.227 | 0.276 | 0.227 | 0.275 |
| TEMP | AVG | **0.161** | **0.309** | 0.187 | 0.338 | 0.295 | 0.413 | 0.302 | 0.423 | 0.315 | 0.442 | 0.302 | 0.423 | 0.292 | 0.418 | 0.291 | 0.406 |
| WIND | AVG | **0.921** | **0.694** | 0.940 | 0.709 | 1.014 | 0.737 | 1.044 | 0.758 | 1.023 | 0.746 | 1.025 | 0.748 | 1.018 | 0.739 | 0.968 | 0.703 |

\* INDICATES FORMER

capture system topology.

**w/o RLC**: Removing structured constraints led to higher test errors, demonstrating that orthogonality and consistency constraints effectively curb overfitting and enforce the learning of intrinsic laws.

**w/o Hierarchy**: Using single-level resolution caused substantial performance decay in long sequences ($L = 720$), validating the effectiveness of the hierarchical decoupling mechanism in shortening long-range dependency paths.

### 3.4. Hyperparameter Analysis

This section systematically investigates the sensitivity of the PESD-TSF framework to two pivotal hyperparameters governing model constraints and prior integration: the RLC regularization weight $\lambda$ and the periodic coefficient $\gamma$.

First, regarding the regularization weight $\lambda$, we evaluated the MSE across the logarithmic scale $[10^{-5}, 10^{-1}]$ (Figure 4(a)). The results uncover a scale-dependent sensitivity: smaller datasets (e.g., ETTh1, ETTh2) manifest a distinct "U-shaped" error curve, indicating a critical need for balanced constraint strength, whereas larger datasets (e.g., Traffic, Electricity) exhibit greater robustness with flatter trajectories. Despite these variations, all datasets consistently achieve minimal error near $\lambda = 10^{-3}$. We observe that

while unconstrained settings ($\lambda \to 0$) risk overfitting, excessive penalties ($\lambda > 10^{-2}$) overly constrict the latent space, limiting feature flexibility. Thus, $\lambda = 10^{-3}$ is identified as the optimal equilibrium for injecting physical inductive bias without hindering expressiveness.

Second, concerning the periodic coefficient $\gamma$, experiments on Traffic and ETTh1 (Figure 4(b)) reveal a convex performance trend within $[0, 1]$. Values that are too low ($\gamma < 0.1$) fail to effectively leverage periodic priors, while values approaching 1.0 ($\gamma > 0.9$) overemphasize periodicity, risking the suppression of underlying trends or the fitting of high-frequency noise. Fortunately, performance remains stable within the moderate range of $[0.3, 0.7]$. Consequently, we adopt a default of $\gamma = 0.5$ to strike a synergistic balance between local periodic feature extraction and global temporal modeling.

### 3.5. Interpretability Analysis: Visualization of Learned Spatiotemporal Dependencies

To validate CSCA's spatial modeling, we compare the physical topology with the learned attention matrix on PeMS04 (Figure 5; methodologies in Appendix G). High diagonal activations align with physical sparsity, confirming that PESD-TSF preserves local neighborhood structures. Moreover, the heatmap reveals denser connectivity: distinct vertical stripes

*Table 3.* Short-term sequence prediction performance of PESD-TSF. The best and second-best results are highlighted in bold and underlined, respectively.

| MODELS | | OURS PESD-TSF | BASELINES TimeMixer (2024b) | SCINet (2022a) | Crossformer (2023) | PatchTST (2023) | TimesNet (2023) | MICN (2022) | DLinear (2023) | DUET (2025) | Stationary (2022b) | Autoformer (2021) | Informer (2021) |
|---|---|---|---|---|---|---|---|---|---|---|---|---|---|
| PeMS03 | MAE | **14.48** | 14.63 | 15.97 | 15.64 | 18.95 | 16.41 | 15.71 | 19.70 | 15.57 | 17.64 | 18.08 | 19.19 |
| | MAPE | **13.64** | 14.54 | 15.89 | 15.74 | 17.29 | 15.17 | 15.67 | 18.35 | 15.27 | 17.56 | 18.75 | 19.58 |
| | RMSE | 23.19 | 23.28 | 25.20 | 25.56 | 30.15 | 26.72 | 24.55 | 32.35 | **22.99** | 28.37 | 27.82 | 32.70 |
| PeMS04 | MAE | 19.41 | **19.21** | 20.35 | 20.38 | 24.86 | 21.63 | 21.62 | 24.62 | 20.84 | 22.34 | 25.00 | 22.05 |
| | MAPE | **11.62** | 12.53 | 12.84 | 12.84 | 16.65 | 13.15 | 13.53 | 16.12 | 14.88 | 14.85 | 16.70 | 14.88 |
| | RMSE | 31.73 | **30.92** | 32.31 | 32.41 | 40.46 | 34.90 | 34.39 | 39.51 | 31.41 | 35.47 | 38.02 | 36.20 |
| PeMS07 | MAE | **19.60** | 20.57 | 22.79 | 22.54 | 27.87 | 25.12 | 22.28 | 28.65 | 22.34 | 26.02 | 26.92 | 27.26 |
| | MAPE | **8.14** | 8.62 | 9.41 | 9.38 | 12.69 | 10.60 | 9.57 | 12.15 | 9.92 | 11.75 | 11.83 | 11.63 |
| | RMSE | **32.72** | 33.59 | 35.61 | 35.49 | 42.56 | 40.71 | 35.40 | 45.02 | 34.97 | 42.34 | 40.60 | 45.81 |
| PeMS08 | MAE | **14.25** | 15.22 | 17.38 | 17.56 | 20.35 | 19.01 | 17.76 | 20.26 | 15.54 | 19.29 | 20.47 | 20.96 |
| | MAPE | **8.87** | 9.67 | 10.80 | 10.92 | 13.15 | 11.83 | 10.76 | 12.09 | 9.87 | 12.21 | 12.27 | 13.20 |
| | RMSE | **23.58** | 24.26 | 27.34 | 27.21 | 31.04 | 30.65 | 27.26 | 32.38 | 24.04 | 38.62 | 31.52 | 30.61 |

*Table 4.* Ablation studies of PESD-TSF on ETT datasets. Performance is measured by MSE and MAE.

| DATASET | PESD-TSF (FULL) MSE | MAE | W/O PERIOD MSE | MAE | W/O RLC MSE | MAE | W/O CSCA MSE | MAE | W/O HIERARCHY MSE | MAE |
|---|---|---|---|---|---|---|---|---|---|---|
| ETTh1 | **0.394** | **0.421** | 0.430 | 0.452 | 0.405 | 0.432 | 0.415 | 0.438 | 0.422 | 0.450 |
| ETTh2 | **0.339** | **0.378** | 0.423 | 0.424 | 0.350 | 0.397 | 0.391 | 0.386 | 0.409 | 0.422 |
| ETTm1 | **0.343** | **0.370** | 0.392 | 0.439 | 0.401 | 0.380 | 0.367 | 0.407 | 0.379 | 0.419 |
| ETTm2 | **0.244** | **0.304** | 0.298 | 0.365 | 0.256 | 0.325 | 0.268 | 0.348 | 0.284 | 0.362 |

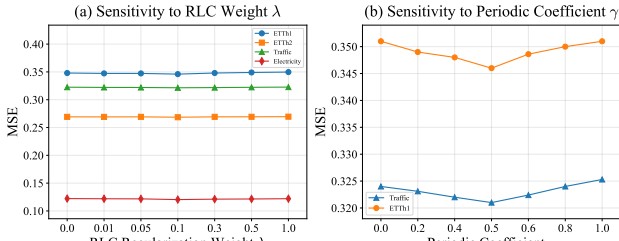

*Figure 4.* Hyperparameter sensitivity analysis on representative datasets. (a) RLC regularization weight $\lambda$: $\lambda \approx 10^{-3}$ yields the lowest MSE, effectively balancing structural constraints with feature flexibility. (b) Periodic coefficient $\gamma$: The "U-shaped" trend indicates an optimal $\gamma \approx 0.5$, effectively integrating periodic priors without introducing noise.

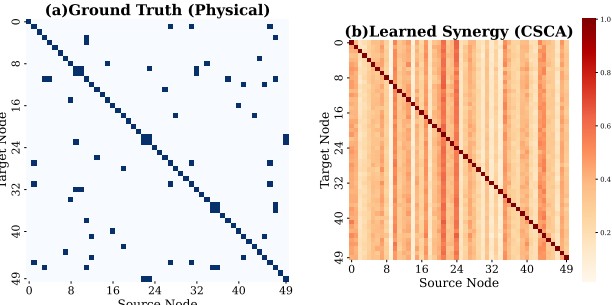

*Figure 5.* Comparison of spatial dependencies on PeMS04. (a) Physical ground truth shows local sparse connectivity; (b) CSCA learned matrix displays vertical stripe patterns.

(e.g., indices 50, 150) identify global "traffic hubs," demonstrating the capture of dynamic long-range dependencies beyond static GNNs. Quantitatively, we evaluate the Top-5 Topology Matching Rate, defined as the Intersection over Union (IoU) between the top-5 edges with the highest attention weights and the ground-truth physical connections. The model achieves a rate of 20.92% (significantly exceeding random $< 1\%$), validating physical interpretability. Regarding the diagonal structure in Figure 5, since the residual connections in our architecture implicitly handle node self-preservation, we superimpose an identity matrix onto the learned cross-variable attention map to represent these in-

herent self-loops. The resulting visualization confirms that PESD-TSF preserves both local neighborhood structures and global traffic hubs.

### 3.6. Spectral Verification of Explicit Decomposition

To verify the physical semantics of PESD-TSF's explicit decomposition, we performed spectral density analysis on the components extracted by IntAttention. As shown in Figure 6(a) (ECL), the trend component is highly concentrated in the low-frequency range ($f < 0.1$), effectively capturing long-term trends, while the variation component

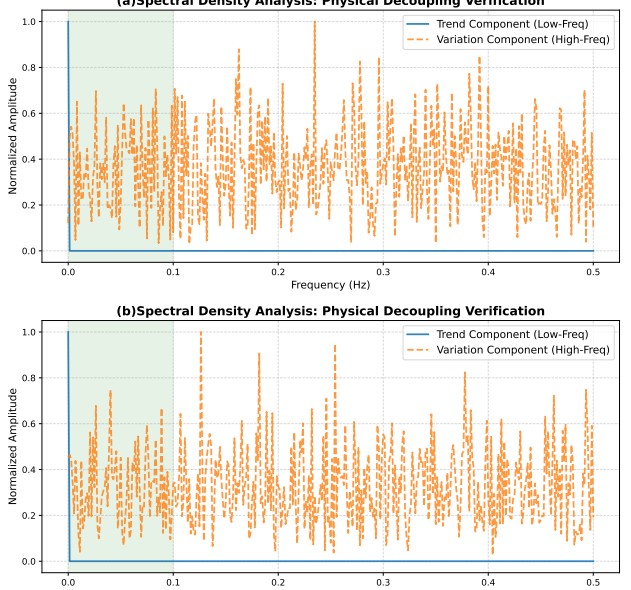

*Figure 6.* Spectral verification on (a) ECL and (b) ETTh1. Trend (blue) concentrates in low frequencies, while variation (orange) dominates medium-to-high frequencies. This separation validates PESD-TSF's physical decoupling.

dominates medium-to-high frequencies ($f > 0.2$). Figure 6(b) (ETTh1) demonstrates even sharper separation, with trends confined to low frequencies and variations exhibiting a broad-band distribution. This consistency confirms that PESD-TSF correctly assigns low frequencies to trends and medium-to-high frequencies to variations. Such adaptive decoupling validates the model's physical inductive bias, minimizing reliance on spurious correlations and significantly enhancing forecasting robustness.

### 3.7. Empirical Analysis of Latent Space Decoupling

To evaluate the decoupling capability of PESD-TSF in the statistical latent space, we conducted a detailed analysis of the orthogonal constraint dynamics on the Solar dataset with a prediction length of $O = 720$.

As illustrated in Figure 7, the orthogonal loss function $\mathcal{L}_{\text{orth}}$ declines sharply during initial training (first 500 steps) and converges to a negligible magnitude ($< 10^{-6}$), confirming that the RLC module effectively enforces orthogonality on the projection basis. Subsequent low-amplitude oscillations indicate that the model maintains orthogonality while optimizing the primary task, balancing accuracy and independence. Figure 7 also visualizes the Pearson correlation matrix of the projected latent factors. Excluding diagonal entries, all off-diagonal values are 0.00, indicating eliminated correlations. This observation aligns with the stability analysis in Appendix B, confirming that PESD-TSF disen-

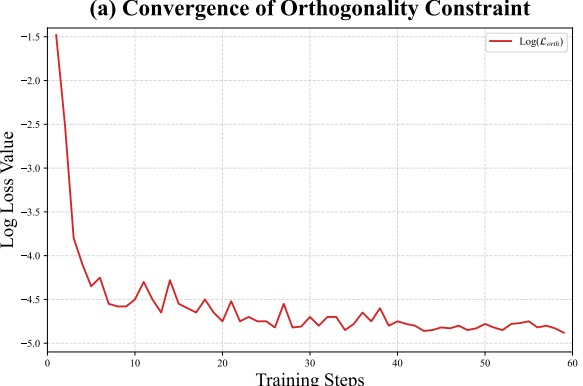

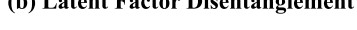

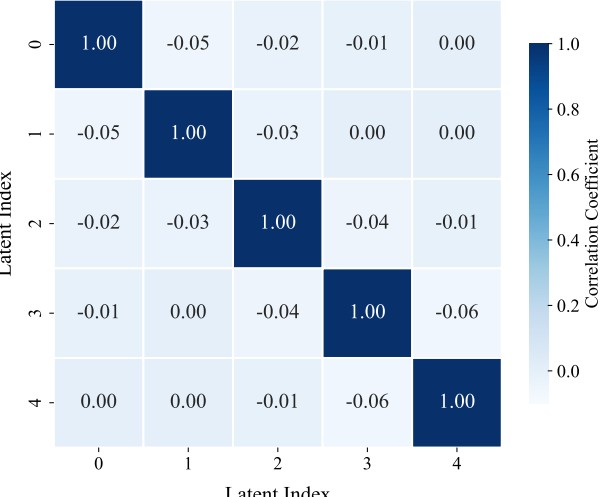

*Figure 7.* RLC analysis on Solar dataset. (a) Orthogonal loss: Rapid decline of $\mathcal{L}_{\text{orth}}$ confirms effective constraint enforcement. (b) Correlation heatmap: Near-zero off-diagonal elements verify effective decoupling of temporal dynamics.

tangles statistical variations into independent orthogonal components. By eliminating redundancy, each factor captures unique temporal patterns, enhancing interpretability and robustness.

### 3.8. Further Analysis

We use additional experiments to examine whether the observed gains are robust to stronger baselines and practical deployment constraints, while leaving more fine-grained sensitivity analyses to Appendix F. To ensure that the improvements do not only arise from comparisons with linear baselines, we add representative attention-based forecasters, including iTransformer, PatchTST, and TimeXer, under the same input length $L = 720$. Table 5 reports the average results over four prediction horizons. PESD-TSF consistently

*Table 5.* Average forecasting performance against strong Transformer baselines under $L = 720$. Results are averaged over $O \in \{96, 192, 336, 720\}$ and reported as MSE/MAE.

| DATASET | PESD-TSF | BEST TRANSFORMER |
|---|---|---|
| TRAFFIC | **0.354/0.242** | 0.380/0.267 (ITRANS.) |
| ELECTRICITY | **0.150/0.244** | 0.161/0.255 (PATCHTST) |
| ETTH1 | **0.394/0.422** | 0.417/0.438 (PATCHTST) |

*Table 6.* Efficiency comparison on Mobility with 5,826 variables. All results are measured on a single GPU under identical settings.

| MODEL | PARAMS (M) | MEM (GB) | TIME (S/EPOCH) |
|---|---|---|---|
| PESD-TSF | 2.29 | **8.37** | **15.56** |
| ITRANSFORMER | **0.80** | 19.30 | 30.76 |

outperforms the strongest Transformer baseline on Traffic, Electricity, and ETTh1, showing that the proposed structured decomposition remains beneficial when compared with competitive attention-based alternatives.

We further examine the practical overhead of PESD-TSF on the high-dimensional Mobility dataset with 5,826 variables. As shown in Table 6, although PESD-TSF has more parameters than iTransformer, it requires substantially less peak memory and shorter training time per epoch. This indicates that CSCA avoids direct full-sequence cross-variable attention: after temporal aggregation, collaboration is modeled mainly along the variable dimension, leading to a favorable efficiency–performance trade-off in high-dimensional scenarios.

### 3.9. Structural Order and Generality

We also test whether the three-stage order is merely a stack of interchangeable modules. Under $L = 720, O = 720$, the full order $S1 \rightarrow S2 \rightarrow S3$ achieves $0.437/0.463$ on ETTh1 and $0.376/0.257$ on Traffic. Reordering the stages degrades performance to $0.448/0.471$ and $0.390/0.266$, while bypassing Stage 2 yields $0.452/0.474$ and $0.394/0.269$. Removing CSCA also increases the errors to $0.447/0.469$ and $0.388/0.265$. These results support the design choice that PESD-TSF first extracts denoised temporal structure, then compresses multi-scale context, and only afterwards reconstructs cross-variable collaboration from time-aggregated representations.

This ordering is consistent with the intended decomposition path of PESD-TSF: temporal regularities are first stabilized by period-aware modulation, local and multi-scale variations are then aggregated through hierarchical processing, and cross-variable collaboration is reconstructed after the temporal representation has been compressed. The design is therefore not simply an accumulation of modules, but a con-

strained modeling sequence that reduces interference among trend, perturbation, and dependency learning. Additional evidence on weakly periodic datasets, the RLC latent dimension, topology matching, and band-wise spectral energy is provided in Appendix F; these results further support the use of structured priors without over-claiming strict physical recovery in every dataset.

Taken together, the supplementary experiments clarify the scope of the proposed model. The Transformer-baseline comparison verifies competitiveness against strong sequence models, the Mobility experiment confirms that cross-variable modeling remains feasible when the number of variables is very large, and the stage-order ablation shows that the performance gain depends on the proposed decomposition order rather than on parameter count alone. These results complement the main forecasting tables and suggest that PESD-TSF is most useful when temporal regularity and inter-variable coupling must be modeled jointly.

## 4. Conclusion

We propose PESD-TSF, a structured decomposition framework that jointly models trend, perturbation, and cross-variable collaboration for long-term forecasting. By integrating periodic gating, a multi-scale structured encoder, CSCA, and RLC regularization, PESD-TSF embeds physical priors directly into the forecasting process. Experiments across diverse benchmarks show strong accuracy, scalability, and interpretable latent structure.

## Acknowledgements

This work was supported in part by the following: the National Natural Science Foundation of China under Grant Nos. U24A20328, U24A20219, 62272281, the Youth Innovation Technology Project of Higher School in Shandong Province under Grant No. 2023KJ212, and the Yantai Natural Science Foundation under Grant No. 2024JCYJ034.

## Impact Statement

This paper presents work whose goal is to advance the field of Machine Learning. There are many potential societal consequences of our work, none which we feel must be specifically highlighted here.

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

# A. Related Works

## A.1. Attention-Based Multi-Scale Time Series Forecasting

Attention-based models remain one of the most influential paradigms for long-term time series modeling (Ni et al., 2023). However, conventional single-scale Transformers suffer from inherent limitations in long-range dependency preservation and periodic structure representation. Informer (Zhou et al., 2021) reduces the computational cost of global attention through the ProbSparse mechanism, enabling larger effective temporal windows, yet it still relies on downstream attention layers to implicitly discover periodic patterns. Autoformer (Wu et al., 2021) replaces dot-product attention with an autocorrelation mechanism, allowing the model to directly capture repetitive temporal segments and thereby enhancing trend-aware representations. TimesNet (Wu et al., 2023) introduces multi-period convolutional blocks to explicitly unfold multiple temporal variations (Liu et al., 2023), preventing periodic structures from being implicitly encoded within attention matrices. More recently, TimeMixer (Wang et al., 2024) further develops decomposable mixing modules to jointly decompose and model multi-scale components within a unified framework, achieving improved scale adaptivity. Web-scale forecasting studies further highlight the importance of scalable model design under heterogeneous temporal signals (Wang et al., 2026a).

Overall, multi-scale attention-based models have evolved from "searching for patterns via global attention scales" (Lu et al., 2025) toward "explicitly constructing decomposable, multi-period structures." Nevertheless, the structural correlations among variables and between trend components remains insufficiently modeled in a unified manner. In contrast, PESD-TSF not only follows the paradigm of explicit decomposition but advances it through a Multi-Scale Structured Encoder, which leverages the CSCA module to reconstruct cross-variable collaborative structures in deep feature spaces that are overlooked by prior methods, enabling unified modeling of multi-scale temporal dynamics and spatial topological dependencies.

## A.2. Utilization of Temporal Priors and Calendar Features

Long-term time series forecasting relies not only on numerical patterns in historical observations but also on structured exogenous priors inherent to real-world systems, such as work cycles, holiday rhythms, and diurnal variations—collectively referred to as calendar-driven factors. DeepAR (Salinas et al., 2020) was among the earliest approaches to integrate temporal prior encodings with sequence modeling, leveraging handcrafted features such as minute-of-hour, day-of-week, and holiday indicators to improve seasonal forecasting performance. N-BEATS (Oreshkin et al., 2019) explicitly incorporates temporal priors into the representation space through trend and seasonality basis functions, reducing the burden on deep networks to implicitly learn periodic structures. Temporal Fusion Transformer (Lim et al., 2021) further combines temporal features, categorical variables, and attention mechanisms, enabling interpretable conditional forecasting guided by external priors. More recently, Uni2TS maps temporal priors into learnable representations, allowing adaptive prior transfer across datasets.

Overall, this research direction has evolved from handcrafted calendar features toward learnable continuous-time embeddings and structured prior encoding (Zhang et al., 2026b;c). However, a unified framework that jointly models multi-period temporal patterns, cross-variable collaboration, and trend-level interaction structure remains underexplored (Wang et al., 2025). To this end, PESD-TSF introduces a Multiplicative Periodic Gating mechanism. Unlike additive positional encodings or simple feature concatenation, this mechanism transforms discrete calendar features into continuous physical priors and applies dynamic amplitude modulation, explicitly enhancing the deep network's ability to perceive and preserve multi-frequency periodic patterns.

## A.3. Latent Factors and Structured Decomposition

Structured decomposition has emerged as a central paradigm for addressing the three-dimensional structure of long-term time series forecasting (Chen et al., 2023; Qiu et al., 2025a), namely trend, perturbation, and cross-variable dependencies. Autoformer (Wu et al., 2021) introduces a learnable residual decomposition that explicitly models trend and seasonality, endowing Transformer architectures with interpretable sequence dynamics. FEDformer (Zhou et al., 2022) incorporates Fourier sparse blocks that project dominant periodic structures into the frequency domain, allowing trend and seasonality to be embedded in latent spaces with low-rank and sparse representations. MICN (Wang et al., 2023) further combines decomposition with mixed convolutional kernels, enabling hierarchical extraction of multi-period components. In parallel, other approaches emphasize the orthogonality and decomposability of latent factors; for instance, ETSformer (Woo et al., 2022) decomposes time series into multiple interpretable latent trajectories, facilitating structural reconstruction in the representation space.

Overall, the evolution of decomposition-based methods has shifted from post hoc component separation toward embedding

structural priors along the forward modeling path. However, most existing approaches focus on either trend or periodic components in isolation and lack a unified, interpretable structural constraint that jointly captures trend dynamics, local perturbations, and cross-variable dependencies interactions. To address this limitation, PESD-TSF introduces a Regularized Latent Component (RLC) module. By enforcing orthogonality and physical consistency constraints, RLC further disentangles the latent space. into three mutually exclusive dimensions—trend, perturbation, and collaboration—thereby providing a mathematical guarantee that the model not only fits the data but also reconstructs dynamical structures consistent with physical intuition.

Recent studies in other structured perception tasks have also emphasized robust representation calibration under ambiguous or compositional inputs, including evidence-driven anchor calibration (Li et al., 2026a), progressive robust learning (Li et al., 2026b), noise mitigation (Chen et al., 2026), uncertainty-aware disambiguation (Chen et al., 2025), modification-frequency balancing (Qiu et al., 2026), and structured attention for medical segmentation (Zhang et al., 2026d). Although these works target retrieval or segmentation rather than forecasting, they reflect a broader trend toward explicitly structured and uncertainty-aware representation learning, which is conceptually aligned with our motivation to reduce entanglement and improve robustness in temporal latent spaces.

## B. Analysis of Orthogonal Projection Stability

We aim to quantify the stability of the RLC projection head by analyzing the response of the latent factors $Z_{rlc} = F_{stat}W_{rlc}$ to perturbations in the statistical feature space. The stability of this linear map is determined by the spectral norm (i.e., the maximum singular value) of its transformation matrix $W_{rlc}$.

Consider a perturbation $\epsilon \in \mathbb{R}^{B \times 2C}$ introduced to the statistical features $F_{stat}$. The resulting variation in the projected latent space is given by:

$$\Delta Z_{rlc} = (F_{stat} + \epsilon)W_{rlc} - F_{stat}W_{rlc} = \epsilon W_{rlc} \tag{29}$$

To measure the magnitude of this variation, we examine its Frobenius norm. Utilizing the consistency property of matrix norms (i.e., $\|\mathbf{AB}\|_F \leq \|\mathbf{A}\|_F \|\mathbf{B}\|_2$), we establish the following upper bound:

$$\|\Delta Z_{rlc}\|_F = \|\epsilon W_{rlc}\|_F \leq \|\epsilon\|_F \|W_{rlc}\|_2 \tag{30}$$

where $\|W_{rlc}\|_2$ denotes the spectral norm of the weight matrix.

During model training, minimizing the orthogonality loss $\mathcal{L}_{orth} = \|W_{rlc}^T W_{rlc} - I\|_F^2$ encourages $W_{rlc}$ to satisfy the column-orthonormality condition $W_{rlc}^T W_{rlc} = I_K$. From a linear algebra perspective, this condition strictly constrains all non-zero singular values of $W_{rlc}$ to be 1. Consequently, as the loss converges, the spectral norm approaches unity:

$$\|W_{rlc}\|_2 = \sigma_{max}(W_{rlc}) \approx 1 \tag{31}$$

Substituting this spectral norm value back into the inequality, we derive the stability bound:

$$\|\Delta Z_{rlc}\|_F \leq \|\epsilon\|_F \cdot \underbrace{\|W_{rlc}\|_2}_{\approx 1} \approx \|\epsilon\|_F \tag{32}$$

This derivation demonstrates that the linear projection from the statistical readout $F_{stat}$ to the RLC latent space satisfies 1-Lipschitz continuity with respect to the projection weights. This property guarantees that the Euclidean distance between any two samples in the statistical feature space is not expanded by the projection head. Theoretically, this acts as a hard constraint on the readout mechanism, ensuring that the auxiliary task remains numerically stable and does not introduce artificial variance into the gradient backpropagation process.

## C. Experiment Setup

### Description

We follow standard LTSF benchmark protocols, consistent with recent state-of-the-art models such as TimeBase and TimeBridge, using a lookback window of 720. Data splits and preprocessing strictly follow the settings adopted by prior benchmark methods.

*Table 7.* Dataset statistics. Var: variables, Length: total length, L: lookback window, O: prediction horizon, Freq: frequency, Scale: total data points.

| | DATASET | VAR | LENGTH | $L$ | $O$ | FREQ | SCALE |
|---|---|---|---|---|---|---|---|
| NORMAL | ETTh1 | 7 | 14,400 | 720 | $96 \sim 720$ | 1h | 0.1M |
| | ETTh2 | 7 | 14,400 | 720 | $96 \sim 720$ | 1h | 0.1M |
| | ETTm1 | 7 | 57,600 | 720 | $96 \sim 720$ | 15m | 0.4M |
| | ETTm2 | 7 | 57,600 | 720 | $96 \sim 720$ | 15m | 0.4M |
| | Weather | 21 | 52,696 | 720 | $96 \sim 720$ | 10m | 1.1M |
| | Electricity | 321 | 26,304 | 720 | $96 \sim 720$ | 1h | 8.1M |
| | Traffic | 862 | 17,544 | 720 | $96 \sim 720$ | 1h | 15.0M |
| | Solar | 137 | 52,560 | 720 | $96 \sim 720$ | 10m | 7.2M |
| | Wind | 7 | 48,673 | 720 | $96 \sim 720$ | 15m | 0.4M |
| | METR-LA | 207 | 34,272 | 720 | $96 \sim 720$ | 5m | 7.1M |
| | AQshunyi | 11 | 35,064 | 720 | $96 \sim 720$ | 1h | 0.4M |
| | AQWan | 11 | 35,064 | 720 | $96 \sim 720$ | 1h | 0.4M |
| | ZafNoo | 11 | 19,225 | 720 | $96 \sim 720$ | 30m | 0.2M |
| | CzeLan | 11 | 19,934 | 720 | $96 \sim 720$ | 30m | 0.2M |
| | PM2.5 | 184 | 11,688 | 720 | $96 \sim 720$ | 3h | 2.2M |
| | Temp | 184 | 11,688 | 720 | $96 \sim 720$ | 3h | 2.2M |
| LARGE | Meter | 2,898 | 26,499 | 1008 | 336 | 30m | 76.8M |
| | Atec | 1,569 | 6,915 | 1008 | 336 | 10m | 10.8M |
| | Mobility | 5,826 | 921 | 35 | 7 | 1d | 5.4M |

## Datasets

We conducted extensive experiments on 20 real-world datasets, categorized into three distinct groups based on their statistical characteristics and scale:

Classic Multivariate Benchmarks: This category comprises seven standard datasets spanning the energy, meteorology, and traffic domains (ETTh1-2, ETTm1-2, Electricity, Weather, Traffic). Serving as the primary reference for SOTA comparisons, these datasets are used to evaluate the model's fundamental capability in fitting non-stationary trends and seasonality.

Diverse Domain-Specific Datasets: We extend our evaluation to include environmental (AQShunyi, AQWan), traffic (Meter-LA), and medical (ZafNoo, CzeLan) domains. Characterized by complex noise distributions and latent dependencies, these datasets are employed to verify the generalization robustness of the model across diverse application scenarios.

High-Dimensional & Strongly Coupled Datasets: We introduce three high-dimensional multivariate datasets: Meter, Mobility, and Atec. These datasets are distinguished by their extreme dimensionality (e.g., Mobility contains 5,826 variables, and Meter contains 2,898) and intense cross-variable dependencies (with average correlation coefficients exceeding 0.84). Spanning cloud computing, energy, and social flow domains, these datasets serve to challenge the computational efficiency of the model when processing thousands of input variables and to validate its ability to capture Global Synergy structures in strongly coupled environments.The statistics of dataset is summarized in Table 7.

## Baseline

We compare PESD-TSF with a comprehensive set of SOTA baselines covering diverse paradigms: **Recent SOTA (2024-2025):** TimeBase (Huang et al., 2025), SparseTSF (Lin et al., 2025), TimeBridge (Liu et al., 2025), Duet (Qiu et al., 2025b), DeformableTST (Luo & Wang, 2024a), U-CAST (Ni et al., 2025), and FITS (Xu et al., 2023). **Transformers:** iTransformer (Liu et al., 2024), PatchTST (Nie et al., 2023), Crossformer (Zhang & Yan, 2023), Non-stationary Transformer (Stationary) (Liu et al., 2022b), Autoformer (Wu et al., 2021), and Informer (Zhou et al., 2021). **Linear, CNN-based &**

**Others:** TimeMixer (Wang et al., 2024), DLinear (Zeng et al., 2023), TimesNet (Wu et al., 2023), TSMixer (Chen et al., 2023), SCINet (Liu et al., 2022a), and MICN (Wang et al., 2023).

**Implementation Details**

All models are implemented in PyTorch and trained on NVIDIA RTX 4060 (8GB) and A800 (80GB) GPUs using the Adam optimizer. The learning rate is selected via grid search from $\{10^{-3}, 10^{-4}, 5 \times 10^{-4}\}$.

To ensure the statistical robustness and reproducibility of our results, each experiment was repeated three times using different random seeds (e.g., 2021–2025). Please refer to Appendix J for specific experimental details.

**Specific Hyperparameters:** To ensure reproducibility, we specify the key structural parameters:

- **Patching:** Patch length $P = 16$ and stride $S = 8$.

- **Dimensions:** Embedding dimension $D = 64$; RLC latent factor count $K = 2C$ (matching the input feature dimension).

- **Coefficients:** Periodic gating intensity $\gamma = 0.5$ (default).

**Baseline Protocol:** Following standard benchmark protocols (Qiu et al., 2024), we fix the lookback window at $L = 720$ for all models to ensure a fair comparison on long-term dependency modeling capabilities, unless a baseline strictly requires a specific input length (e.g., rigid architecture constraints), in which case we follow its official configuration.

## D. Metrics

### D.1. Long-term Forecasting

We adopt Mean Squared Error (MSE) and Mean Absolute Error (MAE) as evaluation metrics for long-term forecasting. Given the ground-truth observations $Y_i$ and the corresponding model predictions $\hat{Y}_i$, the metrics are defined as:

$$\text{MSE} = \frac{1}{N_{\text{eval}}} \sum_{i=1}^{N_{\text{eval}}} (Y_i - \hat{Y}_i)^2, \quad \text{MAE} = \frac{1}{N_{\text{eval}}} \sum_{i=1}^{N_{\text{eval}}} |Y_i - \hat{Y}_i|,$$

where $N_{\text{eval}}$ denotes the total number of predicted values.

### D.2. Short-term Forecasting

For short-term forecasting, we evaluate performance using MAE (as defined above), Mean Absolute Percentage Error (MAPE), and Root Mean Squared Error (RMSE), defined as:

$$\text{MAPE} = \frac{1}{N_{\text{eval}}} \sum_{i=1}^{N_{\text{eval}}} \left| \frac{Y_i - \hat{Y}_i}{Y_i} \right| \times 100, \quad \text{RMSE} = \sqrt{\frac{1}{N_{\text{eval}}} \sum_{i=1}^{N_{\text{eval}}} (Y_i - \hat{Y}_i)^2}.$$

## E. Full Results

### E.1. Main Experiments

Tables 8, 9, and 10 present the comprehensive forecasting results derived from our experiments. Specifically, Table 8 details performance on **Classic Benchmarks** (Table 1), covering seven standard datasets including ETT, Electricity, and Traffic, while Table 9 extends the evaluation to **Diverse Domain-Specific Datasets** (Table 2), encompassing Solar, Air Quality, and Medical domains (Luo & Wang, 2024a). Additionally, Table 10 reports results on **High-Dimensional Benchmarks**. Regarding the lookback window setting: for the datasets in Tables 8 and 9, the input length was fixed at 720 to strictly evaluate long-range modeling capabilities. In contrast, for the high-dimensional datasets in Table 10, we adopted specific protocols: Meter and Atec utilize an input length of 1008 to forecast 336 steps ($3\times$ horizon), whereas Mobility employs an input length of 35 to forecast 7 steps ($5\times$ horizon).

### E.2. Ablation Studies

Table 11 presents the full results of the ablation studies discussed in the main text.

*Table 8.* Long-term sequence prediction performance of PESD-TSF. All results are averaged across four different prediction lengths: O ∈ {96, 192, 336, 720}. The best and second-best results are highlighted in bold and underlined, respectively.

| | | Ours | | 2025($l_{WEIGHT}$) | | 2024($l_{WEIGHT}$) | | 2025 | | 2023-2024 | | | | | | | |
|---|---|---|---|---|---|---|---|---|---|---|---|---|---|---|---|---|---|
| MODELS | O | PESD-TSF | | TIMEBASE | | SPARSETSF | | TIMEBRIDGE | | iTRANS* | | DEFORM** | | TIMEMIXER | | PATCHTST | |
| | | MSE | MAE | MSE | MAE | MSE | MAE | MSE | MAE | MSE | MAE | MSE | MAE | MSE | MAE | MSE | MAE |
| ETTm1 | 96 | 0.281 | 0.334 | 0.311 | 0.351 | 0.314 | 0.359 | 0.284 | 0.337 | 0.300 | 0.353 | 0.291 | 0.347 | 0.293 | 0.345 | 0.293 | 0.346 |
| | 192 | 0.321 | 0.336 | 0.338 | 0.371 | 0.348 | 0.376 | 0.317 | 0.367 | 0.345 | 0.382 | 0.325 | 0.372 | 0.335 | 0.372 | 0.333 | 0.370 |
| | 336 | 0.360 | 0.393 | 0.364 | 0.386 | 0.368 | 0.386 | 0.361 | 0.394 | 0.374 | 0.398 | 0.359 | 0.390 | 0.368 | 0.386 | 0.369 | 0.369 |
| | 720 | 0.410 | 0.420 | 0.413 | 0.414 | 0.419 | 0.413 | 0.413 | 0.418 | 0.429 | 0.430 | 0.418 | 0.423 | 0.426 | 0.417 | 0.416 | 0.420 |
| | Avg | 0.343 | 0.370 | 0.357 | 0.381 | 0.362 | 0.384 | 0.344 | 0.379 | 0.362 | 0.391 | 0.348 | 0.383 | 0.355 | 0.380 | 0.353 | 0.382 |
| ETTm2 | 96 | 0.156 | 0.242 | 0.162 | 0.256 | 0.167 | 0.259 | 0.157 | 0.243 | 0.175 | 0.266 | 0.169 | 0.258 | 0.165 | 0.256 | 0.166 | 0.256 |
| | 192 | 0.214 | 0.283 | 0.218 | 0.293 | 0.219 | 0.297 | 0.217 | 0.285 | 0.242 | 0.312 | 0.229 | 0.299 | 0.225 | 0.298 | 0.223 | 0.296 |
| | 336 | 0.267 | 0.320 | 0.270 | 0.328 | 0.271 | 0.330 | 0.269 | 0.321 | 0.282 | 0.340 | 0.280 | 0.333 | 0.277 | 0.332 | 0.274 | 0.329 |
| | 720 | 0.340 | 0.374 | 0.352 | 0.380 | 0.353 | 0.380 | 0.348 | 0.378 | 0.378 | 0.398 | 0.349 | 0.384 | 0.360 | 0.387 | 0.362 | 0.385 |
| | Avg | 0.244 | 0.304 | 0.251 | 0.314 | 0.253 | 0.317 | 0.246 | 0.310 | 0.269 | 0.329 | 0.257 | 0.319 | 0.257 | 0.318 | 0.256 | 0.317 |
| ETTh1 | 96 | 0.346 | 0.384 | 0.349 | 0.384 | 0.362 | 0.389 | 0.350 | 0.389 | 0.386 | 0.405 | 0.369 | 0.396 | 0.372 | 0.401 | 0.370 | 0.400 |
| | 192 | 0.386 | 0.412 | 0.387 | 0.410 | 0.404 | 0.412 | 0.388 | 0.414 | 0.424 | 0.440 | 0.410 | 0.417 | 0.413 | 0.430 | 0.413 | 0.429 |
| | 336 | 0.407 | 0.428 | 0.408 | 0.418 | 0.435 | 0.428 | 0.408 | 0.430 | 0.449 | 0.460 | 0.391 | 0.414 | 0.438 | 0.450 | 0.422 | 0.440 |
| | 720 | 0.437 | 0.463 | 0.439 | 0.446 | 0.426 | 0.448 | 0.443 | 0.463 | 0.495 | 0.487 | 0.447 | 0.464 | 0.486 | 0.484 | 0.447 | 0.468 |
| | Avg | 0.394 | 0.421 | 0.396 | 0.415 | 0.407 | 0.419 | 0.397 | 0.424 | 0.439 | 0.448 | 0.404 | 0.423 | 0.427 | 0.441 | 0.413 | 0.434 |
| ETTh2 | 96 | 0.268 | 0.329 | 0.292 | 0.345 | 0.294 | 0.346 | 0.271 | 0.331 | 0.297 | 0.348 | 0.272 | 0.334 | 0.281 | 0.351 | 0.274 | 0.337 |
| | 192 | 0.333 | 0.370 | 0.339 | 0.387 | 0.340 | 0.377 | 0.335 | 0.370 | 0.371 | 0.403 | 0.325 | 0.369 | 0.349 | 0.387 | 0.314 | 0.382 |
| | 336 | 0.357 | 0.385 | 0.358 | 0.410 | 0.360 | 0.398 | 0.371 | 0.402 | 0.404 | 0.428 | 0.319 | 0.373 | 0.366 | 0.413 | 0.329 | 0.384 |
| | 720 | 0.398 | 0.430 | 0.400 | 0.448 | 0.383 | 0.425 | 0.387 | 0.425 | 0.424 | 0.444 | 0.395 | 0.433 | 0.401 | 0.436 | 0.379 | 0.422 |
| | Avg | 0.339 | 0.378 | 0.347 | 0.398 | 0.344 | 0.387 | 0.341 | 0.382 | 0.374 | 0.406 | 0.328 | 0.377 | 0.349 | 0.397 | 0.324 | 0.381 |
| WTH | 96 | 0.141 | 0.182 | 0.146 | 0.198 | 0.174 | 0.231 | 0.144 | 0.184 | 0.159 | 0.208 | 0.146 | 0.198 | 0.147 | 0.198 | 0.149 | 0.198 |
| | 192 | 0.186 | 0.227 | 0.185 | 0.241 | 0.216 | 0.267 | 0.186 | 0.225 | 0.200 | 0.248 | 0.191 | 0.239 | 0.192 | 0.243 | 0.194 | 0.241 |
| | 336 | 0.236 | 0.265 | 0.236 | 0.281 | 0.260 | 0.299 | 0.237 | 0.267 | 0.253 | 0.289 | 0.241 | 0.280 | 0.247 | 0.284 | 0.245 | 0.282 |
| | 720 | 0.307 | 0.318 | 0.309 | 0.331 | 0.325 | 0.345 | 0.307 | 0.320 | 0.321 | 0.338 | 0.310 | 0.331 | 0.318 | 0.330 | 0.314 | 0.334 |
| | Avg | 0.217 | 0.248 | 0.219 | 0.263 | 0.244 | 0.286 | 0.219 | 0.249 | 0.233 | 0.271 | 0.222 | 0.262 | 0.226 | 0.264 | 0.226 | 0.264 |
| TRAFFIC | 96 | 0.321 | 0.223 | 0.394 | 0.267 | 0.389 | 0.268 | 0.340 | 0.240 | 0.363 | 0.265 | 0.355 | 0.261 | 0.369 | 0.257 | 0.360 | 0.249 |
| | 192 | 0.348 | 0.240 | 0.403 | 0.271 | 0.399 | 0.272 | 0.343 | 0.250 | 0.385 | 0.273 | 0.380 | 0.271 | 0.400 | 0.272 | 0.379 | 0.256 |
| | 336 | 0.372 | 0.249 | 0.417 | 0.278 | 0.417 | 0.279 | 0.363 | 0.257 | 0.396 | 0.277 | 0.393 | 0.281 | 0.407 | 0.272 | 0.392 | 0.264 |
| | 720 | 0.376 | 0.257 | 0.456 | 0.298 | 0.449 | 0.299 | 0.393 | 0.271 | 0.445 | 0.312 | 0.434 | 0.300 | 0.461 | 0.316 | 0.432 | 0.286 |
| | Avg | 0.354 | 0.242 | 0.418 | 0.279 | 0.414 | 0.280 | 0.360 | 0.255 | 0.397 | 0.282 | 0.391 | 0.278 | 0.409 | 0.279 | 0.391 | 0.264 |
| ECL | 96 | 0.121 | 0.214 | 0.139 | 0.231 | 0.139 | 0.239 | 0.120 | 0.214 | 0.138 | 0.237 | 0.132 | 0.234 | 0.153 | 0.256 | 0.129 | 0.222 |
| | 192 | 0.144 | 0.237 | 0.153 | 0.245 | 0.155 | 0.250 | 0.142 | 0.237 | 0.157 | 0.256 | 0.148 | 0.248 | 0.168 | 0.269 | 0.147 | 0.240 |
| | 336 | 0.163 | 0.258 | 0.169 | 0.262 | 0.171 | 0.260 | 0.156 | 0.252 | 0.167 | 0.264 | 0.165 | 0.266 | 0.189 | 0.291 | 0.163 | 0.259 |
| | 720 | 0.173 | 0.269 | 0.207 | 0.294 | 0.208 | 0.300 | 0.179 | 0.278 | 0.194 | 0.286 | 0.197 | 0.296 | 0.228 | 0.320 | 0.197 | 0.290 |
| | Avg | 0.150 | 0.244 | 0.167 | 0.258 | 0.168 | 0.264 | 0.149 | 0.245 | 0.164 | 0.261 | 0.161 | 0.261 | 0.185 | 0.284 | 0.159 | 0.253 |

\* INDICATES FORMER; \*\* INDICATES ABLETST

# F. Additional Analysis

## F.1. Weakly Periodic Datasets

We examine whether PESD-TSF remains effective when periodic patterns are weak or not visually obvious. Table 12 compares the full model with a variant without Periodic Gating on PM2.5, AQShunyi, and CzeLan. Removing the periodic module increases the error on all three datasets, but the degradation is moderate compared with strongly seasonal datasets. This behavior is consistent with the role of Periodic Gating as an adaptive prior rather than a hard assumption: when calendar-driven structure is weak, the gain of the periodic pathway naturally decreases, while multi-scale decomposition and cross-variable collaboration continue to provide robust forecasting capacity.

## F.2. RLC Latent Dimension

We analyze the latent dimension $K$ in the RLC module, which controls the trade-off between compression and statistical decorrelation. As shown in Table 13, overly small $K$ values produce low latent correlation but sacrifice forecasting accuracy due to excessive bottlenecking. In contrast, overly large $K$ values preserve more information but increase redundancy and weaken the regularization effect. The setting $K = 2C$ achieves the best MSE/MAE on ETTh1, suggesting a balanced configuration between information preservation and latent-space decorrelation.

*Table 9.* Long-term sequence prediction performance of PESD-TSF. All results are averaged across four different prediction lengths: O ∈ {96, 192, 336, 720}. The best and second-best results are highlighted in bold and underlined, respectively.

| | | OURS | | BASELINES | | | | | | | | | | | | |
|---|---|---|---|---|---|---|---|---|---|---|---|---|---|---|---|---|
| **MODELS** | O | PESD-TSF | | TIMEBASE | | SPARSETSF | | TIMEMIXER | | FITS | | iTRANS* | | DLINEAR | | PATCHTST | |
| | | MSE | MAE | MSE | MAE | MSE | MAE | MSE | MAE | MSE | MAE | MSE | MAE | MSE | MAE | MSE | MAE |
| AQSHUNYI | 96 | **0.621** | **0.461** | 0.629 | 0.489 | 0.751 | 0.553 | 0.703 | 0.534 | 0.759 | 0.555 | 0.686 | 0.508 | 0.751 | 0.579 | 0.643 | 0.476 |
| | 192 | **0.654** | **0.481** | 0.661 | 0.500 | 0.774 | 0.551 | 0.720 | 0.524 | 0.774 | 0.553 | 0.728 | 0.520 | 0.771 | 0.571 | 0.692 | 0.501 |
| | 336 | **0.684** | **0.503** | **0.683** | 0.510 | 0.752 | 0.544 | 0.747 | 0.546 | 0.755 | 0.544 | 0.732 | 0.529 | 0.758 | 0.580 | 0.695 | 0.514 |
| | 720 | **0.704** | **0.507** | 0.725 | 0.523 | 0.763 | 0.537 | 0.772 | 0.541 | 0.763 | 0.538 | 0.744 | 0.524 | 0.748 | 0.559 | 0.732 | 0.520 |
| | AVG | **0.665** | **0.488** | 0.675 | 0.506 | 0.760 | 0.546 | 0.736 | 0.536 | 0.763 | 0.548 | 0.723 | 0.520 | 0.757 | 0.572 | 0.691 | 0.503 |
| AQWAN | 96 | **0.711** | **0.451** | 0.726 | 0.477 | 0.758 | 0.486 | 0.762 | 0.484 | 0.795 | 0.510 | 0.798 | 0.511 | 0.835 | 0.531 | 0.735 | 0.467 |
| | 192 | **0.760** | **0.478** | 0.761 | 0.492 | 0.851 | 0.542 | 0.779 | 0.492 | 0.765 | 0.485 | 0.785 | 0.501 | 0.884 | 0.558 | 0.792 | 0.500 |
| | 336 | **0.793** | **0.494** | **0.783** | 0.502 | 0.810 | 0.531 | 0.857 | 0.559 | 0.861 | 0.564 | 0.818 | 0.537 | 0.842 | 0.550 | 0.804 | 0.525 |
| | 720 | **0.813** | **0.497** | 0.847 | 0.524 | 0.883 | 0.546 | 0.891 | 0.549 | 0.831 | 0.517 | 0.971 | 0.604 | 0.972 | 0.600 | 0.909 | 0.561 |
| | AVG | **0.769** | **0.480** | 0.779 | 0.499 | 0.826 | 0.526 | 0.822 | 0.521 | 0.813 | 0.519 | 0.843 | 0.538 | 0.883 | 0.560 | 0.810 | 0.513 |
| CZELAN | 96 | **0.161** | **0.202** | 0.177 | 0.224 | 0.204 | 0.263 | 0.182 | 0.233 | 0.208 | 0.268 | 0.202 | 0.259 | 0.187 | 0.239 | 0.184 | 0.236 |
| | 192 | **0.201** | **0.237** | 0.211 | 0.250 | 0.227 | 0.274 | 0.216 | 0.259 | 0.242 | 0.292 | 0.233 | 0.282 | 0.226 | 0.271 | 0.214 | 0.257 |
| | 336 | **0.225** | **0.261** | 0.243 | 0.276 | 0.239 | 0.305 | 0.246 | 0.312 | 0.247 | 0.316 | 0.239 | 0.305 | 0.238 | 0.301 | 0.229 | 0.290 |
| | 720 | **0.249** | **0.286** | 0.269 | 0.297 | 0.292 | 0.325 | 0.291 | 0.322 | 0.303 | 0.338 | 0.305 | 0.340 | 0.315 | 0.349 | 0.288 | 0.319 |
| | AVG | **0.209** | **0.246** | 0.225 | 0.262 | 0.241 | 0.292 | 0.234 | 0.282 | 0.250 | 0.304 | 0.245 | 0.297 | 0.242 | 0.290 | 0.229 | 0.276 |
| ZAFNOO | 96 | 0.429 | **0.381** | 0.435 | 0.407 | 0.435 | 0.439 | **0.410** | 0.411 | 0.458 | 0.461 | 0.442 | 0.447 | 0.423 | 0.423 | 0.412 | 0.413 |
| | 192 | 0.484 | **0.420** | 0.470 | 0.435 | 0.548 | 0.509 | 0.473 | 0.437 | 0.506 | 0.468 | 0.560 | 0.522 | 0.534 | 0.494 | 0.518 | 0.479 |
| | 336 | 0.526 | **0.464** | **0.511** | 0.453 | 0.525 | 0.474 | 0.576 | 0.520 | 0.528 | 0.477 | 0.536 | 0.486 | 0.596 | 0.537 | 0.528 | 0.477 |
| | 720 | 0.569 | **0.478** | 0.565 | 0.484 | 0.617 | 0.540 | **0.560** | 0.487 | 0.590 | 0.515 | 0.623 | 0.546 | 0.606 | 0.527 | 0.581 | 0.506 |
| | AVG | 0.502 | **0.435** | **0.495** | 0.445 | 0.531 | 0.491 | 0.505 | 0.464 | 0.521 | 0.480 | 0.540 | 0.500 | 0.540 | 0.495 | 0.510 | 0.469 |
| METR-LA | 96 | **1.013** | **0.577** | 1.028 | 0.655 | 1.056 | 0.667 | 1.051 | 0.657 | 1.047 | 0.661 | 1.023 | 0.640 | 1.096 | 0.685 | 1.053 | 0.658 |
| | 192 | **1.142** | **0.625** | 1.112 | 0.711 | 1.232 | 0.730 | 1.163 | 0.689 | 1.295 | 0.772 | 1.196 | 0.713 | 1.330 | 0.788 | 1.134 | 0.672 |
| | 336 | **1.243** | **0.663** | 1.317 | 0.771 | 1.356 | 0.772 | 1.304 | 0.737 | 1.309 | 0.743 | 1.320 | 0.750 | 1.353 | 0.764 | 1.250 | 0.706 |
| | 720 | **1.328** | **0.764** | 1.558 | 0.877 | 1.562 | 0.885 | 1.564 | 0.879 | 1.534 | 0.865 | 1.474 | 0.834 | 1.423 | 0.800 | 1.404 | 0.789 |
| | AVG | **1.181** | **0.657** | 1.254 | 0.754 | 1.302 | 0.764 | 1.271 | 0.741 | 1.296 | 0.760 | 1.253 | 0.734 | 1.301 | 0.759 | 1.210 | 0.706 |
| PM2.5 | 96 | **0.422** | **0.430** | 0.432 | 0.435 | 0.475 | 0.482 | 0.445 | 0.449 | 0.495 | 0.501 | 0.466 | 0.474 | 0.437 | 0.441 | 0.450 | 0.454 |
| | 192 | **0.420** | **0.420** | 0.429 | 0.432 | 0.458 | 0.465 | 0.479 | 0.483 | 0.455 | 0.460 | 0.505 | 0.511 | 0.454 | 0.458 | 0.461 | 0.465 |
| | 336 | **0.387** | **0.403** | 0.426 | 0.433 | 0.478 | 0.482 | 0.473 | 0.475 | 0.450 | 0.453 | 0.461 | 0.467 | 0.449 | 0.451 | 0.475 | 0.477 |
| | 720 | **0.319** | **0.375** | 0.425 | 0.448 | 0.456 | 0.479 | 0.464 | 0.487 | 0.440 | 0.461 | 0.489 | 0.514 | 0.457 | 0.479 | 0.448 | 0.470 |
| | AVG | **0.387** | **0.407** | 0.428 | 0.437 | 0.467 | 0.477 | 0.465 | 0.474 | 0.460 | 0.469 | 0.480 | 0.492 | 0.449 | 0.457 | 0.459 | 0.467 |
| SOLAR | 96 | **0.153** | **0.193** | 0.192 | 0.239 | 0.205 | 0.241 | 0.232 | 0.271 | 0.218 | 0.257 | 0.217 | 0.255 | 0.200 | 0.234 | 0.205 | 0.239 |
| | 192 | **0.182** | **0.216** | 0.213 | 0.252 | 0.215 | 0.265 | 0.238 | 0.293 | 0.207 | 0.256 | 0.208 | 0.257 | 0.239 | 0.294 | 0.227 | 0.280 |
| | 336 | **0.194** | **0.227** | 0.222 | 0.261 | 0.213 | 0.276 | 0.234 | 0.301 | 0.229 | 0.295 | 0.238 | 0.309 | 0.231 | 0.298 | 0.225 | 0.290 |
| | 720 | **0.201** | **0.236** | 0.235 | 0.264 | 0.232 | 0.272 | 0.273 | 0.319 | 0.261 | 0.307 | 0.270 | 0.319 | 0.237 | 0.277 | 0.249 | 0.291 |
| | AVG | **0.182** | **0.218** | 0.216 | 0.254 | 0.216 | 0.264 | 0.244 | 0.296 | 0.229 | 0.279 | 0.233 | 0.285 | 0.227 | 0.276 | 0.227 | 0.275 |
| TEMP | 96 | **0.133** | **0.281** | 0.142 | 0.286 | 0.149 | 0.300 | 0.158 | 0.316 | 0.168 | 0.338 | 0.155 | 0.310 | 0.165 | 0.329 | 0.145 | 0.291 |
| | 192 | **0.131** | **0.279** | 0.155 | 0.305 | 0.168 | 0.323 | 0.163 | 0.312 | 0.181 | 0.351 | 0.168 | 0.324 | 0.174 | 0.335 | 0.166 | 0.319 |
| | 336 | **0.132** | **0.282** | 0.176 | 0.323 | 0.220 | 0.373 | 0.249 | 0.421 | 0.223 | 0.381 | 0.232 | 0.395 | 0.246 | 0.417 | 0.221 | 0.374 |
| | 720 | **0.251** | **0.396** | 0.274 | 0.437 | 0.642 | 0.654 | 0.637 | 0.643 | 0.689 | 0.699 | 0.652 | 0.663 | 0.584 | 0.590 | 0.632 | 0.639 |
| | AVG | **0.161** | **0.309** | 0.187 | 0.338 | 0.295 | 0.413 | 0.302 | 0.423 | 0.315 | 0.442 | 0.302 | 0.423 | 0.292 | 0.418 | 0.291 | 0.406 |
| WIND | 96 | **0.757** | 0.598 | 0.785 | 0.626 | 0.796 | 0.589 | 0.908 | 0.669 | 0.787 | **0.581** | 0.824 | 0.609 | 0.824 | 0.607 | 0.802 | 0.591 |
| | 192 | **0.885** | 0.672 | 0.902 | 0.690 | 0.949 | 0.698 | 1.040 | 0.764 | 0.910 | 0.674 | 1.054 | 0.779 | 0.961 | 0.707 | 0.944 | 0.694 |
| | 336 | **0.977** | 0.728 | 0.997 | 0.737 | 1.182 | 0.851 | 1.039 | 0.747 | 1.100 | 0.795 | 1.025 | 0.742 | 1.022 | 0.734 | 1.043 | 0.750 |
| | 720 | **1.068** | 0.780 | 1.075 | 0.782 | 1.127 | 0.808 | 1.189 | 0.851 | 1.295 | 0.932 | 1.196 | 0.863 | 1.266 | 0.907 | 1.084 | **0.776** |
| | AVG | **0.921** | **0.694** | 0.940 | 0.709 | 1.014 | 0.737 | 1.044 | 0.758 | 1.023 | 0.746 | 1.025 | 0.748 | 1.018 | 0.739 | 0.968 | 0.703 |

\* INDICATES FORMER

## F.3. Topology and Frequency Statistics

We further refine the interpretation of the topology and frequency analyses with quantitative evidence. For the learned cross-variable dependency map on PeMS04, PESD-TSF obtains an off-diagonal Top-10 matching rate of $8.42\%$ against the physical adjacency, compared with $1.69\%$ for random selection, indicating statistically meaningful alignment with known topology rather than full mechanistic recovery. For spectral decomposition, the trend component allocates $62.57/24.73/12.70\%$ of its energy to low/mid/high frequency bands, whereas the variation component allocates $28.99/33.64/37.37\%$. These results support a calibrated claim: PESD-TSF concentrates trend information in low frequencies and distributes variation more broadly over mid-to-high frequencies, while avoiding overstatement of strict physical interpretability.

## G. Physical Topology Verification of Cross-Variable Synergy (CSCA)

### G.1. Physical Ground Truth Construction

The physical topology ground truth is derived from the sensor distance matrix provided by the PeMS datasets. To address the sparsity and directionality of the raw data, we implemented the following preprocessing steps to construct a robust

*Table 10.* Performance benchmarking of PESD-TSF on high-dimensional multivariate time series with intense cross-variable dependencies. The best and second-best results are highlighted in bold and underlined, respectively.

| | | OURS | | BASELINES | | | | | | | | | | | | |
|---|---|---|---|---|---|---|---|---|---|---|---|---|---|---|---|---|
| MODELS | | PESD-TSF | | U-CAST | | DUET | | TIMESNET | | TSMIXER | | ITRANSFORMER | | DLINEAR | | PATCHTST | |
| DATASET | O | MSE | MAE | MSE | MAE | MSE | MAE | MSE | MAE | MSE | MAE | MSE | MAE | MSE | MAE | MSE | MAE |
| METER | AVG | **0.936** | **0.533** | 0.943 | 0.551 | 1.308 | 0.731 | 1.034 | 0.586 | 0.987 | 0.564 | 0.949 | 0.556 | 0.944 | 0.549 | 1.254 | 0.706 |
| ATEC | AVG | **0.287** | **0.267** | **0.287** | 0.280 | 0.330 | 0.339 | 0.493 | 0.429 | 0.398 | 0.387 | 0.345 | 0.319 | 0.318 | 0.314 | 0.298 | 0.298 |
| MOBILITY | AVG | **0.299** | **0.300** | 0.315 | 0.317 | 0.439 | 0.410 | 0.410 | 0.388 | 1.165 | 0.787 | 0.312 | 0.314 | 0.344 | 0.359 | 0.344 | 0.341 |

evaluation benchmark:

- **Symmetrization:** Considering the bidirectional nature of traffic flow influence (e.g., congestion waves propagate both upstream and downstream), we symmetrized the distance matrix such that $A_{ij} = A_{ji}$.

- **2-Hop Expansion:** Given the extreme sparsity of the PeMS04 road network (where first-order connections are minimal), we extended the physical neighborhood to include second-order neighbors (i.e., "neighbors of neighbors") to comprehensively evaluate connectivity. This resulted in a connectivity-based binary adjacency matrix.

### G.2. Attention Map Extraction and Aggregation

The PESD-TSF model employs a multi-head cross-scale attention mechanism. To extract a unified dependency matrix from the model for visualization, we performed the following aggregation operations:

- **Multi-Head Averaging:** We averaged the attention weights across all $H$ heads to obtain a representative global view of the learned dependencies.

- **Cross-Scale Aggregation:** Since the CSCA module captures dependencies across different scales, we spatially aligned and aggregated the attention matrices, projecting them back into the original sensor topology space $\mathbb{R}^{N \times N}$.

- **Self-Loop Representation via Residuals:** Recognizing that the residual connections in the Transformer architecture implicitly preserve a node's own historical information—while the attention mechanism focuses on cross-variable interactions—we incorporate an identity matrix into the final visualization. This operation mathematically represents the inherent self-loops maintained by the residual stream, ensuring the heatmap accurately reflects the complete information flow (both self-preserving and interactive) consistent with the physical ground truth.

## H. Visualization of Explicit Decomposition Results

While PESD-TSF is designed to explicitly decompose time series into semantically distinct components, quantitative forecasting metrics alone do not fully reveal whether the learned representations exhibit the intended temporal behaviors. In particular, numerical errors cannot directly reflect how the extracted components are utilized during the forecasting process in the time domain. To provide further qualitative evidence, we visualize the explicit decomposition results produced by the proposed model.

Specifically, we select a representative variable from the test set of ETTh1 and visualize a contiguous segment of the input sequence together with the corresponding decomposed components generated by PESD-TSF. For clarity and readability, only a single decomposition scale is shown when multi-scale modeling is applied.

Figure 8 presents the visualization results. The extracted trend component exhibits smooth and slowly varying temporal dynamics that closely align with the long-term evolution of the original signal. In contrast, the residual component primarily captures localized fluctuations and transient variations, without introducing spurious long-term structures. Moreover, the predicted future sequence follows the extrapolated trend rather than fitting short-term perturbations, indicating that the proposed structured decomposition guides the model toward stable long-term forecasting behavior.

*Table 11.* Full ablation studies on ETT, Weather, Traffic, Electricity, and Solar datasets. Performance is measured by MSE and MAE.

| | | OURS | | ABLATION VARIANTS | | | | | | | |
| --- | --- | --- | --- | --- | --- | --- | --- | --- | --- | --- | --- |
| | | PESD-TSF | | W/O PERIOD | | W/O RLC | | W/O CSCA | | W/O HIERARCHY | |
| DATASET | O | MSE | MAE | MSE | MAE | MSE | MAE | MSE | MAE | MSE | MAE |
| ETTM1 | 96 | **0.281** | **0.334** | 0.345 | 0.397 | 0.485 | 0.341 | 0.323 | 0.353 | 0.331 | 0.359 |
| | 192 | **0.321** | **0.336** | 0.361 | 0.402 | 0.332 | 0.347 | 0.336 | 0.374 | 0.352 | 0.386 |
| | 336 | **0.360** | **0.393** | 0.418 | 0.448 | 0.371 | 0.402 | 0.396 | 0.429 | 0.409 | 0.446 |
| | 720 | **0.410** | **0.420** | 0.442 | 0.507 | 0.416 | 0.428 | 0.413 | 0.473 | 0.422 | 0.483 |
| | AVG | **0.343** | **0.370** | 0.392 | 0.439 | 0.401 | 0.380 | 0.367 | 0.407 | 0.379 | 0.419 |
| ETTM2 | 96 | **0.156** | **0.242** | 0.198 | 0.312 | 0.168 | 0.270 | 0.182 | 0.304 | 0.193 | 0.307 |
| | 192 | **0.214** | **0.283** | 0.284 | 0.365 | 0.229 | 0.329 | 0.264 | 0.354 | 0.285 | 0.373 |
| | 336 | **0.267** | **0.320** | 0.325 | 0.371 | 0.273 | 0.321 | 0.283 | 0.342 | 0.312 | 0.364 |
| | 720 | **0.340** | **0.374** | 0.386 | 0.410 | 0.353 | 0.381 | 0.342 | 0.390 | 0.344 | 0.402 |
| | AVG | **0.244** | **0.304** | 0.298 | 0.365 | 0.256 | 0.325 | 0.268 | 0.348 | 0.284 | 0.362 |
| ETTH1 | 96 | **0.346** | **0.384** | 0.371 | 0.409 | 0.352 | 0.398 | 0.361 | 0.400 | 0.377 | 0.430 |
| | 192 | **0.386** | **0.412** | 0.412 | 0.424 | 0.394 | 0.420 | 0.394 | 0.413 | 0.421 | 0.444 |
| | 336 | **0.407** | **0.428** | 0.441 | 0.475 | 0.419 | 0.434 | 0.421 | 0.459 | 0.438 | 0.454 |
| | 720 | **0.437** | **0.463** | 0.496 | 0.498 | 0.454 | 0.477 | 0.485 | 0.479 | 0.453 | 0.472 |
| | AVG | **0.394** | **0.421** | 0.430 | 0.452 | 0.405 | 0.432 | 0.415 | 0.438 | 0.422 | 0.450 |
| ETTH2 | 96 | **0.268** | **0.329** | 0.320 | 0.380 | 0.277 | 0.340 | 0.297 | 0.346 | 0.301 | 0.386 |
| | 192 | **0.333** | **0.370** | 0.428 | 0.401 | 0.337 | 0.392 | 0.390 | 0.387 | 0.410 | 0.400 |
| | 336 | **0.357** | **0.385** | 0.461 | 0.428 | 0.379 | 0.406 | 0.431 | 0.392 | 0.472 | 0.461 |
| | 720 | **0.398** | **0.430** | 0.484 | 0.485 | 0.405 | 0.450 | 0.446 | 0.438 | 0.454 | 0.440 |
| | AVG | **0.339** | **0.378** | 0.423 | 0.424 | 0.350 | 0.397 | 0.391 | 0.386 | 0.409 | 0.422 |
| WEATHER | 96 | **0.141** | **0.182** | 0.152 | 0.197 | 0.144 | 0.186 | 0.148 | 0.191 | 0.154 | 0.199 |
| | 192 | **0.186** | **0.227** | 0.200 | 0.245 | 0.189 | 0.231 | 0.195 | 0.238 | 0.203 | 0.248 |
| | 336 | **0.236** | **0.265** | 0.254 | 0.286 | 0.240 | 0.270 | 0.247 | 0.278 | 0.258 | 0.290 |
| | 720 | **0.307** | **0.318** | 0.330 | 0.343 | 0.313 | 0.324 | 0.322 | 0.333 | 0.335 | 0.348 |
| | AVG | **0.217** | **0.248** | 0.234 | 0.268 | 0.222 | 0.253 | 0.228 | 0.260 | 0.238 | 0.271 |
| TRAFFIC | 96 | **0.321** | **0.223** | 0.343 | 0.239 | 0.326 | 0.226 | 0.334 | 0.232 | 0.347 | 0.241 |
| | 192 | **0.348** | **0.240** | 0.372 | 0.257 | 0.353 | 0.244 | 0.362 | 0.250 | 0.376 | 0.259 |
| | 336 | **0.372** | **0.249** | 0.398 | 0.266 | 0.378 | 0.253 | 0.387 | 0.259 | 0.402 | 0.269 |
| | 720 | **0.376** | **0.257** | 0.402 | 0.275 | 0.382 | 0.261 | 0.391 | 0.267 | 0.406 | 0.278 |
| | AVG | **0.354** | **0.242** | 0.379 | 0.259 | 0.360 | 0.246 | 0.369 | 0.252 | 0.383 | 0.262 |
| ELECTRICITY | 96 | **0.121** | **0.214** | 0.132 | 0.233 | 0.123 | 0.218 | 0.127 | 0.225 | 0.131 | 0.231 |
| | 192 | **0.144** | **0.237** | 0.157 | 0.258 | 0.147 | 0.242 | 0.151 | 0.249 | 0.156 | 0.256 |
| | 336 | **0.163** | **0.258** | 0.178 | 0.281 | 0.166 | 0.263 | 0.171 | 0.271 | 0.176 | 0.279 |
| | 720 | **0.173** | **0.269** | 0.189 | 0.293 | 0.176 | 0.274 | 0.182 | 0.282 | 0.187 | 0.291 |
| | AVG | **0.150** | **0.244** | 0.164 | 0.266 | 0.153 | 0.249 | 0.158 | 0.257 | 0.163 | 0.264 |
| SOLAR | 96 | **0.153** | **0.193** | 0.164 | 0.207 | 0.156 | 0.197 | 0.161 | 0.203 | 0.165 | 0.208 |
| | 192 | **0.182** | **0.216** | 0.195 | 0.231 | 0.186 | 0.220 | 0.191 | 0.227 | 0.197 | 0.233 |
| | 336 | **0.194** | **0.227** | 0.208 | 0.243 | 0.198 | 0.232 | 0.204 | 0.238 | 0.210 | 0.245 |
| | 720 | **0.201** | **0.236** | 0.215 | 0.253 | 0.205 | 0.241 | 0.211 | 0.248 | 0.217 | 0.255 |
| | AVG | **0.183** | **0.218** | 0.196 | 0.234 | 0.186 | 0.223 | 0.192 | 0.229 | 0.197 | 0.235 |

## I. Adaptive Gating Behavior Analysis

To further investigate how the proposed periodic gating mechanism responds to varying temporal characteristics, we visualize its activation patterns alongside a proxy measure of signal complexity in the time domain. Quantitative performance metrics alone cannot reveal whether the gating mechanism adapts meaningfully to local temporal variations during forecasting.

Specifically, we select a representative variable from the ETTh1 test set and compute local signal volatility using a sliding-window standard deviation as a measure of temporal complexity. Figure 9 illustrates the raw input signal and its

*Table 12.* Applicability on weakly periodic or non-obviously periodic datasets.

| DATASET | FULL | W/O PERIOD | Δ |
|---|---|---|---|
| PM2.5 | **0.319/0.375** | 0.329/0.383 | +0.010/+0.008 |
| AQSHUNYI | **0.704/0.507** | 0.715/0.515 | +0.011/+0.008 |
| CZELAN | **0.249/0.286** | 0.256/0.292 | +0.007/+0.006 |

*Table 13.* Ablation on latent dimension $K$ in RLC on ETTh1 ($L = 720, O = 720$).

| $K$ | MSE | MAE | LC |
|---|---|---|---|
| $4 \ (\approx 0.5C)$ | 0.482 | 0.508 | **0.038** |
| $7 \ (= C)$ | 0.456 | 0.482 | 0.072 |
| $10 \ (\approx 1.5C)$ | 0.445 | 0.471 | 0.109 |
| $14 \ (= 2C)$ | **0.437** | **0.463** | 0.136 |
| $28 \ (= 4C)$ | 0.441 | 0.468 | 0.254 |

*Table 14.* Performance comparison between PESD-TSF and the runner-up baseline TimeBridge. Results are reported as mean ± standard deviation.

| MODEL | PESD-TSF (OURS) | | TIMEBRIDGE(2025) | |
|---|---|---|---|---|
| DATASET | MSE | MAE | MSE | MAE |
| ETTM1 | $0.343 \pm 0.009$ | $0.370 \pm 0.008$ | $0.344 \pm 0.014$ | $0.379 \pm 0.010$ |
| ETTM2 | $0.244 \pm 0.004$ | $0.304 \pm 0.004$ | $0.246 \pm 0.004$ | $0.310 \pm 0.012$ |
| ETTH1 | $0.394 \pm 0.014$ | $0.421 \pm 0.004$ | $0.397 \pm 0.010$ | $0.424 \pm 0.008$ |
| ETTH2 | $0.339 \pm 0.008$ | $0.378 \pm 0.008$ | $0.341 \pm 0.018$ | $0.382 \pm 0.015$ |
| WEATHER | $0.217 \pm 0.009$ | $0.248 \pm 0.006$ | $0.219 \pm 0.006$ | $0.249 \pm 0.004$ |
| ELECTRICITY | $0.150 \pm 0.011$ | $0.244 \pm 0.014$ | $0.149 \pm 0.011$ | $0.245 \pm 0.007$ |
| TRAFFIC | $0.354 \pm 0.013$ | $0.242 \pm 0.012$ | $0.360 \pm 0.008$ | $0.255 \pm 0.013$ |

corresponding local volatility (top), together with the learned periodic gating intensity over time (bottom).

As shown in Figure 9, periods with higher local volatility and stronger temporal fluctuations are consistently associated with elevated gating responses. In contrast, during relatively stable intervals, the gating intensity remains close to its learned mean value. This alignment indicates that the periodic gating mechanism adaptively modulates its activation strength in response to changing temporal dynamics, rather than behaving as a static or uniformly activated module.

Importantly, this visualization does not imply explicit causal attribution, but provides qualitative evidence that the proposed gating mechanism exhibits structured and interpretable temporal behavior. By selectively emphasizing periodic representations in complex or highly varying regions, the gating mechanism contributes to more stable and robust long-term forecasting.

## J. Statistical Analysis

We repeated all experiments three times and reported the standard deviations for both our model and the second-best baseline, along with the results of statistical significance tests. Table 14, Table 15, and Table 16 present the results for long-term and short-term forecasting, respectively.

## K. Future Directions

Although PESD-TSF has demonstrated strong predictive performance and clear physical interpretability in long-term time series forecasting, its structured modeling paradigm leaves several promising directions for future exploration. Motivated

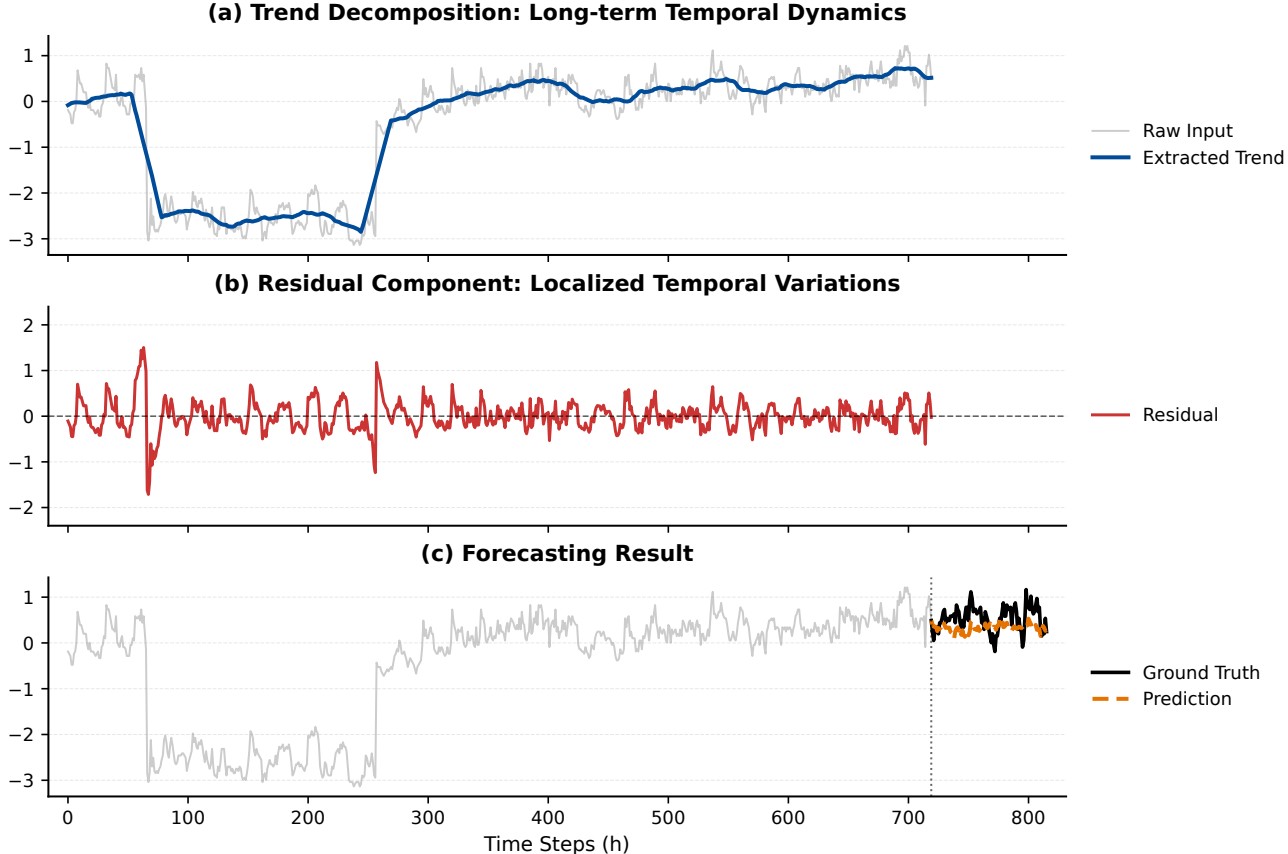

*Figure 8.* Time-domain visualization of explicit decomposition results produced by PESD-TSF on a representative test sample from ETTh1. The figure illustrates the original input sequence, the extracted trend component, the residual component, and the resulting forecasting behavior compared with the ground truth.

*Table 15.* Performance comparison between PESD-TSF and the runner-up baseline TimeBase. Results are reported as mean ± standard deviation.

| MODEL | PESD-TSF (OURS) | | TIMEBASE(2025) | |
| --- | --- | --- | --- | --- |
| DATASET | MSE | MAE | MSE | MAE |
| AQSHUNYI | $0.665 \pm 0.017$ | $0.461 \pm 0.012$ | $0.675 \pm 0.018$ | $0.506 \pm 0.014$ |
| AQWAN | $0.769 \pm 0.012$ | $0.480 \pm 0.010$ | $0.779 \pm 0.014$ | $0.499 \pm 0.012$ |
| CZELAN | $0.209 \pm 0.006$ | $0.246 \pm 0.004$ | $0.225 \pm 0.008$ | $0.262 \pm 0.008$ |
| ZAFNOO | $0.502 \pm 0.007$ | $0.435 \pm 0.008$ | $0.495 \pm 0.007$ | $0.445 \pm 0.009$ |
| METR-LA | $1.181 \pm 0.023$ | $0.657 \pm 0.015$ | $1.254 \pm 0.025$ | $0.754 \pm 0.017$ |
| PM2.5 | $0.387 \pm 0.008$ | $0.407 \pm 0.006$ | $0.428 \pm 0.011$ | $0.437 \pm 0.007$ |
| SOLAR | $0.182 \pm 0.004$ | $0.218 \pm 0.005$ | $0.216 \pm 0.004$ | $0.254 \pm 0.003$ |
| TEMP | $0.161 \pm 0.006$ | $0.309 \pm 0.007$ | $0.187 \pm 0.008$ | $0.338 \pm 0.006$ |
| WIND | $0.921 \pm 0.010$ | $0.694 \pm 0.012$ | $0.940 \pm 0.014$ | $0.709 \pm 0.008$ |

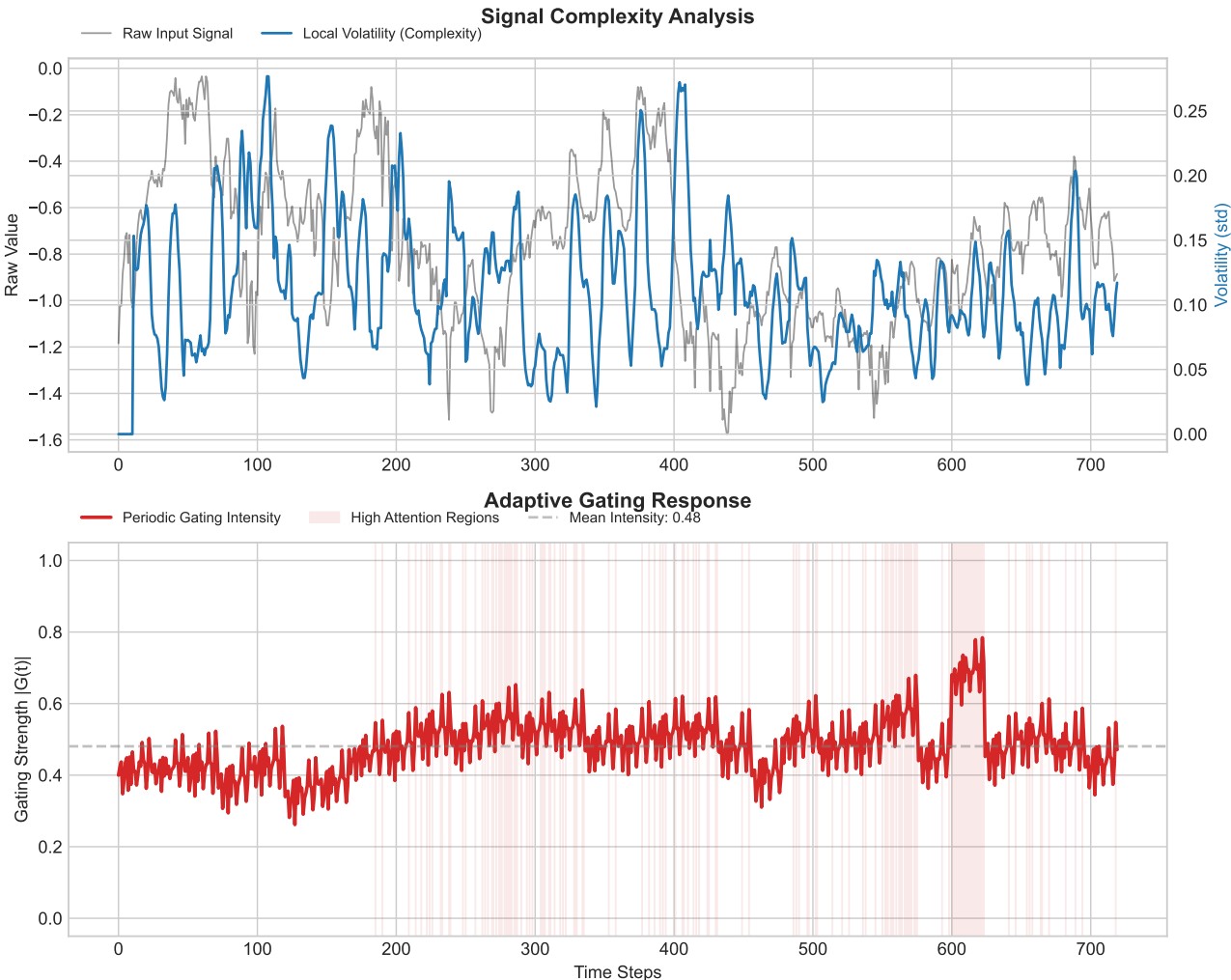

*Figure 9.* Visualization of adaptive periodic gating on a representative ETTh1 test sample. The top panel shows the input signal and local volatility, while the bottom panel presents the corresponding gating intensity. Higher activation aligns with increased temporal complexity.

*Table 16.* Short-term forecasting performance comparison between PESD-TSF and TimeMixer on PeMS datasets. Results are reported as mean ± standard deviation.

| MODEL | PESD-TSF (OURS) | | | TIMEMIXER (2024B) | | |
|---|---|---|---|---|---|---|
| DATASET | MAE | MAPE | RMSE | MAE | MAPE | RMSE |
| PEMS03 | $14.48 \pm 0.142$ | $13.64 \pm 0.123$ | $23.19 \pm 0.165$ | $14.63 \pm 0.112$ | $14.54 \pm 0.105$ | $23.28 \pm 0.128$ |
| PEMS04 | $19.41 \pm 0.142$ | $11.62 \pm 0.107$ | $31.73 \pm 0.114$ | $19.21 \pm 0.217$ | $12.53 \pm 0.154$ | $30.92 \pm 0.143$ |
| PEMS07 | $19.60 \pm 0.166$ | $8.14 \pm 0.154$ | $32.72 \pm 0.188$ | $20.57 \pm 0.158$ | $8.62 \pm 0.112$ | $33.59 \pm 0.273$ |
| PEMS08 | $14.25 \pm 0.276$ | $8.87 \pm 0.121$ | $23.58 \pm 0.145$ | $15.22 \pm 0.311$ | $9.67 \pm 0.101$ | $24.26 \pm 0.212$ |

by our consistent empirical observation that explicitly reconstructing the underlying dynamical structure is often more effective for generalization than merely increasing model scale, a key direction is to extend PESD-TSF toward a pre-trained foundation model for time series analysis. Unlike existing pretraining paradigms that primarily rely on implicit temporal representations, this direction aims to leverage the explicit decoupling of trends, variations, and cross-variable correlations to learn transferable, structure-aware dynamical priors, thereby enabling more robust adaptation in zero-shot or few-shot scenarios.

In addition, for safety-critical applications such as power grid management, where reliable uncertainty estimation is essential, future work will explore integrating probabilistic frameworks—such as normalizing flows or diffusion models—into the RLC latent space. By modeling uncertainty under explicit structural consistency constraints, this extension would allow PESD-TSF to go beyond point forecasts and quantify plausible confidence intervals, providing risk-aware predictions suitable for high-stakes decision-making.

Finally, we aim to relax the discrete-time assumption by extending the periodicity-aware gating mechanism to continuous-time settings. By incorporating Neural Ordinary Differential Equations (Neural ODEs) to model structured dynamics, PESD-TSF could naturally handle irregular sampling and missing observations without relying on interpolation artifacts, further improving its robustness and applicability in real-world temporal systems.

