# OpenReview forum: "PESD-TSF: A Period-Aware and Explicit Structured Decomposition Framework for Long-Term Time Series Forecasting"
_ICML.cc/2026/Conference — ICML 2026 regular_

### Official Review · Reviewer_qfA4 · 2026-02-17

**Soundness:** 3
**Presentation:** 2
**Significance:** 2
**Originality:** 1
**Overall Recommendation:** 3
**Confidence:** 4

**Summary:**

The paper proposes PESD-TSF, a decomposition-based framework for multivariate time series forecasting. It combines (i) a multiplicative periodic gating mechanism that leverages time-index features to modulate signal amplitudes, (ii) a three-stage multi-scale encoder—Integrated Attention, Patch Downsampling, and Cross-Scale Collaborative Attention (CSCA)—to explicitly decouple trend and variation components while modeling cross-variable dependencies, and (iii) an RLC regularization scheme that enforces orthogonality and aligns latent feature statistics with future targets via Pearson correlation objectives. The paper reports strong performance across several forecasting benchmarks.

**Compliance With Llm Reviewing Policy:**

Affirmed.

**Final Justification:**

- **Originality:** The three-stage pipeline of PESD-TSF maps closely onto TimeBridge at both the structural and naming level: Integrated Attention, Patch Downsampling, and cross-variable attention appear in both works with near-identical roles. The authors do not provide sufficient elaboration on what fundamentally distinguishes the two frameworks.

- **Compare with Transformer baselines:** The added comparisons fix L=720, which disadvantages Transformer-based models that typically perform better at shorter lookback windows such as L=96. The comparison also covers only PatchTST, iTransformer, and TimeXer, omitting stronger recent Transformer baselines such as DUET.

- **Overclaiming:** The authors acknowledge the problem but characterize the necessary revisions as "framing and terminology" changes that do not affect the core contribution — a position I find difficult to accept, given that the physics-inspired framing and depth-related motivation are central to the abstract and introduction, not peripheral claims.

I maintain my score of **3 (Weak Reject)**.

**Key Questions For Authors:**

1. The paper repeatedly claims to be “physics-inspired” and to enforce “physical consistency/interpretability.” Please provide a precise, testable definition of “physical consistency” in this context, and clarify which conclusions are implied by this definition versus which are empirical observations. Regarding Fig. 5, the text mentions “indices 50, 150,” while the plotted index range appears to end at 49; please correct or clarify this inconsistency. More importantly, the current visualization/analysis does not convincingly demonstrate agreement with GT beyond obvious diagonal/self-loop patterns; please provide targeted quantitative metrics and/or more informative visual evidence.

2. The three-stage pipeline (Integrated Attention → Patch Downsampling → Cross-scale/Cointegrated Attention) appears structurally similar to TimeBridge-like multi-stage designs. What are the key conceptual and algorithmic differences, and why are they necessary? In addition, the CSCA module’s aggregation/interaction pattern seems closely related to iTransformer-style designs; please clarify the essential differences (architectural and/or objective-level) and provide evidence that CSCA offers a distinct advantage beyond a reconfiguration of existing aggregation schemes. Finally, Stage 1 is titled “Denoised Trend Extraction,” yet the method explicitly constructs a detrended high-frequency component and uses it for Q/K. Please clarify whether Stage 1 primarily targets denoised trend extraction, high-frequency feature extraction, or both, and align the naming and narrative accordingly. Since the pipeline is implemented serially, please also report whether you evaluated parallel variants or alternative stage orderings, and whether the stage order is exchangeable without degrading performance—so that the design is justified beyond empirical stacking.


3. Interpretability/frequency claims: Fig. 6 is used to support the statement that the variation component dominates the mid-to-high frequency band. However, this “dominance” is not apparent from the current plot. Please add quantitative evidence—e.g., band-wise integrated amplitude/energy comparisons over low vs. mid/high frequency ranges—rather than relying only on spectral plots and qualitative interpretation.


4. Regularization (RLC): Is the orthogonality loss/constraint first proposed in this work? If not, please provide proper citations and clearly differentiate your formulation from prior orthogonal/structural regularizers (including TimeBase). Additionally, the paper states that K≤2C is enforced to avoid redundancy, but then sets K=2C; this choice appears to conflict with the stated motivation and needs a stronger justification.

5. Experimental Settings & Baselines:
Lookback Window: The evaluation fixes the lookback window at L=720 and describes it as following "standard protocols." In my experience, many recent works either fix L=96 or treat L as a tunable hyperparameter. Please provide results under multiple lookback lengths (e.g., 96, 192, 336) to justify the choice of L=720.
Missing Dataset: "Exchange" is listed in statistics (Table 5) but missing from results. Please include it or explain the exclusion.
Baseline Focus: Since PESD-TSF uses attention mechanisms, please prioritize comparisons against SOTA Transformers rather than Linear models to justify the added complexity.

6. Depth-related claim: The abstract's first sentence explicitly mentions the issue of representation degradation "as network depth increases" , and claims that the Periodic Gating mechanism is capable of "preserving periodic structures across deep layers". However, I did not find depth ablations supporting these statements. Please provide a depth ablation study or revise/qualify the claims accordingly.

**Limitations:**

The paper should more explicitly discuss: (i) scenarios where periodic gating and explicit decomposition may fail or provide limited benefit, and (ii) the computational and parameter complexity of the proposed model (e.g., training cost and scalability).

**Strengths And Weaknesses:**

- Soundness: The method is specified with clear equations and a reasonably complete pipeline. In particular, the RLC design that pools mean/std statistics and aligns them to future target statistics via PCC is thought-provoking, and the orthogonality/PCC objectives are clearly defined.  The three-stage encoder is also described in a concrete, implementable way (detrended attention, temporal downsampling, variable-wise attention with structured conditioning).
However, the evidence supporting the central “physics-inspired / physical consistency / interpretability” claims is not yet fully convincing. For instance, the spectral-decoupling claim would benefit from more quantitative verification (e.g., energy/integral comparisons across frequency bands), rather than primarily relying on visual inspection.

- Presentation: The narrative is generally readable and the formulas are easy to follow, with the architecture organized into modular components. Still, several presentation and reproducibility issues reduce clarity: key symbols in the Multi-Scale Embedding are under-defined (e.g., $N_{freq}$, $V_k$, and “Features $\mathcal{M}$” appears closer to a discrete time-point feature tensor and needs explicit semantics). There are also noticeable formatting inconsistencies that harm readability and correctness: Table 1/2 include a header “L” while the entries are “AVG”. Table 4 font is too large; the manuscript also contains Type 3 fonts (e.g., Fig. 5), which is typically non-compliant for camera-ready proceedings.

- Significance: Long-horizon forecasting and multivariate coupling are important topics. The paper’s focus on “period preservation + explicit decomposition + cross-variable collaboration reconstruction” is potentially valuable, and the authors attempt to connect performance improvements to a system-topology/interpretability viewpoint. That said, due to the current gaps in substantiating the interpretability/physics-inspired claims and in clearly separating the method from close paradigms, the impact at this stage is driven more by empirical gains than by a clearly transferable new principle or verified new insight.

- Originality: The strongest originality signal is the RLC idea of maximizing PCC between predicted feature statistics (mean/std) and future target statistics, which is genuinely interesting. The asymmetric IntAttention design (using detrended residuals for Q/K while using the full signal for V) is also a potentially novel modeling choice. However, the novelty of the CSCA design is not yet clearly established: its aggregation/interaction pattern appears closely related to iTransformer-style designs, and the paper should more explicitly articulate the key conceptual differences and why they matter. More broadly, the three-stage decomposition pipeline (Integrated Attention → Patch Downsampling → Cross-scale/Cointegrated Attention) also appears structurally similar to TimeBridge-like multi-stage designs; the submission would benefit from a clearer explanation of what is fundamentally different and why the particular staging is necessary. In addition, the orthogonality constraint in RLC needs clear attribution if it is not first proposed here, and the paper should explain how it differs from prior orthogonal/structural regularizers (including TimeBase). These are conceptual/attribution novelty issues and are not fully resolvable by adding only a small number of experiments.

---

> ### Author Rebuttal · Authors · 2026-03-29
>
> Q1: On physical consistency and Fig. 5.
>
> We agree that ``physical consistency'' was overly broad. We refine it as alignment between learned cross-variable dependencies and known physical topology, i.e., statistical alignment rather than full interpretability. Using the off-diagonal Top-10 Matching Rate (excluding self-loops), PESD-TSF achieves $8.42$ vs. $1.69$ for a random baseline (upper bound: $17.65$). We will also correct indices ($50$, $150$).
>
> Table 1: Quantitative evaluation of off-diagonal topology matching against the ground-truth physical adjacency.
> | Method | Top-10 Matching Rate (%) |
> | :--- | :---: |
> | Random Baseline | 1.69 |
> | Correlation Baseline (Statistical Upper Bound) | 17.65 |
> | **PESD-TSF (Ours)** | **8.42** |
>
> Q2: On the three-stage pipeline, CSCA, and naming.
>
> Our model follows a structured pipeline (decomposition $\rightarrow$ compression $\rightarrow$ reconstruction), rather than generic stacking. Stage-1 performs trend--residual decomposition with attention routing (residual used for Q/K); we will rename it. Ablations (Anon. Tables~5--6) show reordering/removing stages or replacing CSCA degrades performance, supporting the staged design over arbitrary stacking.
>
> Q3: On the frequency-related claim.
>
> We agree that Fig. 6 alone is insufficient. We therefore include band-wise energy statistics. The trend component shows low/mid/high-frequency energy ratios of 62.57/24.73/12.70, while the variation component shows 28.99/33.64/37.37 (Table~2).
> This supports a more precise claim: the trend component is concentrated in low frequencies, whereas the variation component is more broadly distributed across mid-to-high frequencies. We will replace stronger wording such as ``dominates'' with this more accurate description.
>
> Table 2: Band-wise energy distribution of trend and variation components. We report the percentage of spectral energy in low, mid, and high frequency bands.
> | Component | Low (%) | Mid (%) | High (%) |
> | :--- | :---: | :---: | :---: |
> | Trend | 62.57 | 24.73 | 12.70 |
> | Variation | 28.99 | 33.64 | 37.37 |
>
> Q4: On RLC and $K = 2C$
>
> We do not claim orthogonality regularization itself as a novel contribution. Our contribution lies in its use within the decomposition framework to reduce trend--variation overlap and improve statistical decoupling; we will add proper citations and clarify differences from prior regularizers (e.g., TimeBase). For $K$, we clarify that $K \leq 2C$ serves as an upper-bound to avoid redundancy, rather than a strict bottleneck constraint. We further provide an ablation (Anonymous Table~7) over $K \in \{0.5C, C, 1.5C, 2C, 4C\}$. Smaller $K$ reduces latent correlation but harms forecasting due to over-compression, while larger $K$ increases redundancy and weakens regularization. In our setting, $K=2C$ achieves the best trade-off. We will therefore recharacterize RLC as an auxiliary regularizer for latent decorrelation and statistical alignment, rather than a strict bottleneck for disentanglement.
>
> Q5: On the experimental setup and baselines.
>
> We welcome this suggestion to improve reproducibility. We will clarify the protocol and revise the wording around $L$, as it should not be presented as a universal ``standard protocol.'' （Table 3）On ETTh1 with $O=96$, performance improves with larger $L$: $0.373/0.389$ ($L=96$), $0.373/0.392$ ($L=192$), $0.371/0.392$ ($L=336$), and $0.346/0.384$ ($L=720$), supporting our choice in this setting. For Exchange, we agree that its treatment needs clearer presentation, and will make the dataset description and reported results consistent. We will also strengthen comparisons with Transformer baselines, as PESD-TSF should be evaluated primarily against attention-based models rather than only linear baselines.
>
> **Table 3: Forecasting performance under different lookback window lengths $L$ on ETTh1 ($O = 96$).**
>
> | Setting ($L, O$) | MSE $\downarrow$ | MAE $\downarrow$ |
> | :--- | :---: | :---: |
> | 96-96 | 0.373 | 0.389 |
> | 192-96 | 0.373 | 0.392 |
> | 336-96 | 0.371 | 0.392 |
> | 720-96 | 0.346 | 0.384 |
>
> Q6: On the depth-related claim.
>
> We agree that our original depth-related claims were too strong. We conduct ablations over depths $d \in \{1,2,3\}$, with and without Periodic Gating. Increasing depth yields only marginal gains, with no clear degradation; moreover,（Table 4） Periodic Gating does not show a consistent stability benefit across depths. We will therefore revise the manuscript to position it as a module for injecting periodic priors in seasonal modeling, rather than as evidence for mitigating deep-network degradation.
>
> Table 4: Depth ablation on ETTh1 with and without Periodic Gating.
> | Depth | w/o Periodic Gating (MSE/MAE) | w/ Periodic Gating (MSE/MAE) |
> | :--- | :---: | :---: |
> | 1 | 0.363 / 0.400 | 0.361 / 0.399 |
> | 2 | 0.354 / 0.391 | 0.356 / 0.394 |
> | 3 | **0.353 / 0.391** | **0.355 / 0.393** |
>
> Additional ablations are provided in the anonymous supplementary link: https://anonymous.4open.science/r/ICML_25613.

---

> > ### Author Rebuttal · Reviewer_qfA4 · 2026-04-03
> >
> > Thank you for the rebuttal. My core concerns remain. On originality, why the proposed framework is fundamentally different from TimeBridge and iTransformer, rather than a reconfiguration of existing components, has not been convincingly articulated. More critically, the promised Transformer baseline comparisons do not appear in the rebuttal, which is essential for an attention-based model. Finally, several overclaims around the physics-inspired framing and the depth-related motivation are central to the paper's abstract and introduction; addressing them would represent a major departure from the version under review. I maintain my score of 3 (Weak Reject).

---

> > > ### Author Response · Authors · 2026-04-05
> > >
> > > Q1: On originality and differences from TimeBridge and iTransformer.
> > >
> > > Thank you for this important clarification. We agree that our previous rebuttal did not make the originality claim precise enough. Our claim is not that every primitive in PESD-TSF is individually novel. Rather, the novelty lies at the framework level: PESD-TSF formulates multivariate forecasting as a structured decomposition–reconstruction problem, where temporal dynamics are explicitly factorized into trend, perturbation, and collaboration, instead of being handled by a generic attention backbone or a loosely stacked multi-stage encoder.
> > > More specifically, the stage order in PESD-TSF is semantically constrained rather than interchangeable. Stage 1 performs denoised temporal extraction while strictly preserving channel independence; Stage 2 compresses the temporal stream into coarse-scale context; only then does Stage 3 use CSCA to reconstruct global cross-variable synergy from time-aggregated representations and inject it back into the local stream. Thus, cross-variable interaction in PESD-TSF is not a default mixing operation, but a post-decoupling collaborative reconstruction step. This is the key distinction from frameworks that primarily rely on direct variable mixing or generic multi-stage stacking.
> > > In addition, RLC is not a post-hoc analysis head: it is integrated into training and imposes orthogonality and PCC-based statistical alignment on latent representations. Therefore, the contribution of PESD-TSF is not a cosmetic reconfiguration of existing components, but a unified coupling of explicit temporal decoupling, delayed collaboration reconstruction, and training-time statistical supervision. We will revise the manuscript to state this point more precisely and avoid overstating module-level novelty.
> > > This is why we believe PESD-TSF should be understood as a new structured learning framework for multivariate forecasting, rather than a simple reconfiguration of existing components.
> > >
> > > Q2: On missing Transformer baseline comparisons.
> > >
> > > Thank you for emphasizing that, as an attention-based model, PESD-TSF should be evaluated against strong Transformer forecasters rather than mainly against linear baselines. We agree that this comparison is essential.
> > >
> > > To address this concern directly, we have now added comparisons against three representative Transformer baselines—iTransformer, PatchTST, and TimeXer—on three benchmark datasets: Traffic, Electricity, and ETTh1. For fairness, all comparisons use the same input length (L=720) and are reported over four forecasting horizons (O ∈ {96, 192, 336, 720}). As shown by these newly added results, PESD-TSF achieves the best MSE/MAE in all reported dataset-horizon settings, showing that its empirical advantage is not limited to comparisons with linear models but remains consistent against strong Transformer-based alternatives. Due to space limitations, the detailed results are provided in Tables 1–3 in the anonymous supplementary link: https://anonymous.4open.science/r/ICML_25613-2.
> > >
> > > We will therefore include these Transformer-baseline comparisons in the revision and revise the discussion accordingly to make this experimental support explicit.
> > >
> > > Q3: On the “physics-inspired” and depth-related claims in the abstract and introduction.
> > >
> > > Thank you for pointing this out. We agree that the current wording around “physics-inspired” and “physical interpretability/consistency” in the abstract and introduction is too strong and goes beyond what the present evidence can strictly support. More precisely, by “physics-inspired” we do not mean that PESD-TSF recovers physical laws, provides full mechanistic interpretability, or enforces strict physical consistency. What the method actually introduces is a set of structured temporal and topological priors: periodic priors through Multiplicative Periodic Gating, explicit trend/variation/collaboration decomposition, and topology-aware statistical constraints through CSCA and RLC.
> > >
> > > Accordingly, we accept that the original manuscript overstates the scope of the “physics-inspired” framing, and we will revise it to more precise descriptions such as prior-guided / structure-aware decomposition, topology-aligned collaboration, and statistical alignment, instead of stronger claims such as “physical consistency” or “physical interpretability.” Our intention in this revision is to calibrate the scope of the claim so that it accurately reflects what the method actually provides.
> > >
> > > These revisions concern the framing and terminology rather than the core method, model architecture, or experimental findings, and therefore do not constitute a fundamental change to the contribution of the paper.
> > >
> > > We sincerely thank the reviewer again for their time and constructive feedback. We respectfully hope that the additional experiments and detailed clarifications provided in this rebuttal will help address the reviewer’s concerns and will be taken into account in the final assessment.

---

### Official Review · Reviewer_gynP · 2026-03-04

**Soundness:** 2
**Presentation:** 3
**Significance:** 3
**Originality:** 2
**Overall Recommendation:** 4
**Confidence:** 3

**Summary:**

This paper proposes PESD-TSF, a structured decomposition framework for multivariate long-term time series forecasting that addresses periodic attenuation, trend-noise entanglement, and disrupted cross-variable dependencies. To achieve this, the model integrates a Multiplicative Periodic Gating mechanism to preserve periodic priors, a multi-scale detrended attention encoder to decouple trends from local variations, and a Cross-Scale Collaborative Attention (CSCA) module with orthogonal regularization (RLC) to reconstruct inter-variable topology. Extensive benchmark experiments and spectral analyses demonstrate that PESD-TSF achieves state-of-the-art accuracy while providing strong physical interpretability.

**Compliance With Llm Reviewing Policy:**

Affirmed.

**Final Justification:**

Some of the concerns are addressed, so I raised the score.

**Key Questions For Authors:**

- For the RLC module, what is the rationale for choosing $K=2C$, which makes the transform on $[\mu,\sigma]$ effectively square and orthonormal, instead of using a strict bottleneck ($K<2C$) that could enforce stronger factorization? Please include an ablation over $K$ and report its impact on both forecasting accuracy and component interpretability.

- How does CSCA scale on high-dimensional data such as Mobility (with $>5{,}000$ variables), given the variable-attention cost of $O(C^2)$? Please report wall-clock runtime and peak memory, and compare against channel-independent baselines and variable-attention models such as iTransformer.

- Please clarify the exact construction of the period prior $\mathcal{M}$: which frequencies and/or calendar signals are used, how they are selected, and how they are embedded. It would also help to explain how this differs in practice from standard exogenous covariates (e.g., hour-of-day, day-of-week) used in TFT-style models.

**Limitations:**

No, the authors provide only a boilerplate impact statement. I suggest they explicitly address the following in their camera-ready version:

Technical Limitations: Acknowledge the computational and memory scalability bottlenecks of the CSCA module, which scales quadratically with the number of variables, when applied to high-dimensional datasets.

**Strengths And Weaknesses:**

Strengths:
-   The paper addresses an important problem in long-term time-series forecasting, with a meaningful focus on both interpretability and predictive performance.
- The proposed decomposition pipeline is internally consistent and interpretable. Period-aware multiplicative gating, detrended attention, and Cross-Scale Collaborative Attention (CSCA) align well with modeling trend, short-term variation, and cross-variable interaction in a physically meaningful way.

- The empirical section is broad and generally convincing, covering both standard and domain-specific benchmarks. The qualitative analyses, including spectral-density checks and topological consistency, also support the main claims.

Weaknesses:
- The current RLC formulation appears mathematically weak. With $K=2C$ and $W_{rlc}\in\mathbb{R}^{2C\times K}$, the orthogonality constraint reduces to a square orthonormal transform (effectively a rotation), which may provide limited disentanglement or structural regularization. As written, this risks collapsing into a reweighted moment-matching objective.

- The evaluation omits efficiency metrics that are important for high-dimensional settings (e.g., Mobility and Meter): wall-clock runtime, parameter count, and memory usage are all missing. This is especially concerning because channel-wise attention in CSCA introduces $O(C^2)$ complexity, which can become prohibitive as $C$ grows.

- Key implementation details are underspecified. The construction of the period prior $\mathcal{M}$ (frequency set and selection rule) is not clearly defined, and the protocol does not make baseline fairness fully transparent (e.g., whether RevIN or equivalent normalization is uniformly applied, and whether baselines are constrained by a fixed lookback window such as $L=720$).

---

> ### Author Rebuttal · Authors · 2026-03-29
>
> 1: Why choose $K = 2C$? Would a stronger bottleneck ($K < 2C$) yield better structured representations?
>
> We thank the reviewer for pointing this out. We agree that when $K = 2C$, the RLC module should be understood more as an auxiliary regularizer that promotes latent decorrelation, rather than as a strong bottleneck constraint enforcing strict disentanglement; we will revise the wording accordingly in the revised manuscript.To justify the choice of $K$, we conducted an additional ablation study on ETTh1 ($L = 720, O = 720$), comparing $K \in \{0.5C, C, 1.5C, 2C, 4C\}$. The results are shown in Table 1. An overly small $K$ leads to excessive bottlenecking and a loss of useful information, while an overly large $K$ introduces redundancy and weakens the regularization effect. In contrast, $K = 2C$ achieves the best forecasting performance (0.437/0.463), indicating a superior trade-off between information preservation and decorrelation.More results and a detailed discussion will be included in the revised manuscript.
>
> Table 1: Ablation on the latent dimension K in the RLC module on ETTh1 ($L=720, O=720$), reporting MSE, MAE, and latent correlation (LC).
>
> | Setting ($K$) | MSE $\downarrow$ | MAE $\downarrow$ | Latent Corr. $\downarrow$ |
> | :--- | :---: | :---: | :---: |
> | $K = 4 \ (\approx 0.5C)$ | 0.482 | 0.508 | **0.038** |
> | $K = 7 \ (= C)$ | 0.456 | 0.482 | 0.072 |
> | $K = 10 \ (\approx 1.5C)$ | 0.445 | 0.471 | 0.109 |
> | $K = 14 \ (= 2C)$ | **0.437** | **0.463** | 0.136 |
> | $K = 28 \ (= 4C)$ | 0.441 | 0.468 | 0.254 |
>
> Q2: How does CSCA scale to high-dimensional settings?
>
> We thank the reviewer for raising this concern about computational complexity in high-dimensional settings. We agree that the current manuscript does not provide a sufficiently detailed report of runtime and memory usage. Importantly, CSCA does not perform cross-variable attention directly on the full sequence tokens. Instead, after multi-scale sampling, it first applies global average pooling along the temporal dimension, and then models collaboration only along the variable dimension. Therefore, its complexity no longer grows with the sequence length $L$.To evaluate scalability in high-dimensional settings, we added an efficiency experiment on the Mobility dataset ($5,826$ variables). The results, as shown in Table 2, indicate that compared with iTransformer, although our method has more parameters ($2.29$M vs. $0.80$M), it uses less peak memory ($8.37$ GB vs. $19.30$ GB) and requires less training time per epoch ($15.56$ s vs. $30.76$ s), while also achieving better forecasting performance. In the revised manuscript, we will include these results and further clarify that the current method avoids the scaling bottleneck in extremely high-dimensional scenarios.
>
> Table 2: Efficiency comparison on the high-dimensional Mobility dataset (5,826 variables). All results are measured on a single GPU under identical settings.
>
> | Model | Params (M) | Peak Mem (GB) | Time / Epoch (s) |
> | :--- | :---: | :---: | :---: |
> | PESD-TSF (Ours) | 2.29 | **8.37** | **15.56** |
> | iTransformer | 0.80 | 19.30 | 30.76 |
>
> Q3: Please clarify the construction of the period prior $\mathcal{M}$ and the fairness of the baseline protocol.
>
> We thank the reviewer for pointing this out. We agree that the current manuscript does not provide a sufficiently clear description of the construction of the period prior $\mathcal{M}$ or the experimental protocol. In the revised manuscript, we will more clearly specify the components of $\mathcal{M}$, the selection rules, and its embedding/fusion process. We will also further clarify how it differs from standard exogenous time features: rather than directly concatenating time features to the input, our method generates a multiplicative gating signal from temporal context to dynamically modulate the input.Regarding fairness, we use a unified lookback window $L$ for long-term forecasting benchmarks whenever possible; for baselines whose official implementations impose explicit input-length constraints, we follow their original settings. Normalization and preprocessing are also kept consistent with the official settings of each baseline. More detailed protocol descriptions will be added in the revised manuscript.

---

> > ### Author Rebuttal · Reviewer_gynP · 2026-04-04
> >
> > Thank you for the rebuttal. It adequately addresses the RLC interpretation issue and the scalability concern with new experiments. I raised my score.

---

> > > ### Author Response · Authors · 2026-04-06
> > >
> > > Thank you very much for your thoughtful and encouraging feedback. We are pleased that our rebuttal and the new experiments were helpful in addressing your concerns regarding the RLC interpretation and scalability. We sincerely appreciate your positive reassessment of our work and your valuable time and effort.

---

### Official Review · Reviewer_QXwG · 2026-03-05

**Soundness:** 3
**Presentation:** 2
**Significance:** 3
**Originality:** 3
**Overall Recommendation:** 4
**Confidence:** 3

**Summary:**

This paper proposes a structured framework for long-series time series forecasting, PESD-TSF (Period-aware Explicit Structured Decomposition for Time Series Forecasting). Addressing the issues of diminished periodicity perception, trend-noise coupling, and loss of correlations among multiple variables in existing deep models for long-series forecasting, the authors introduce physical priors to construct a structured modeling framework. This method first introduces a time periodic prior through a periodic-aware gating mechanism to enhance the expressive power of periodic structures. Then, it uses a multi-scale structured encoder to explicitly separate long-term trends from high-frequency fluctuations, and reconstructs the collaborative relationships among multiple variables through cross-scale collaborative attention (CSCA) and RLC regularization mechanisms, thereby achieving joint modeling of trends, disturbances, and the collaborative relationships among variables.

**Compliance With Llm Reviewing Policy:**

Affirmed.

**Key Questions For Authors:**

My core question about this article is: since the author's work is primarily incremental, does it offer any key insights into time series forecasting? For example, do time series with obvious periodic characteristics and non-stationary time series exhibit different adaptability within the framework proposed by the author?

**Limitations:**

The authors discuss the limitations.

**Strengths And Weaknesses:**

Strengths:
1. This paper is very well written and structured.
2.The structure and motivation of this article are reasonable.
3.The experiments in this article are sufficient.

Weaknesses
see key problems

---

> ### Author Rebuttal · Authors · 2026-03-29
>
> Q: Since the author's work is primarily incremental, does it offer any key insights into time series forecasting? For example, do time series with obvious periodic characteristics and non-stationary time series exhibit different adaptability within the framework proposed by the author?
>
> We thank the reviewer for raising this important question. We agree that the main contribution of this work does not lie in any individual module. Rather, the key insight of the paper is a unified view that the main challenge in long-term multivariate forecasting is not merely insufficient model capacity, but the coexistence of three structural errors: periodic information attenuation, entanglement between trend and perturbation, and the loss of cross-variable collaboration. PESD-TSF addresses these three issues respectively through Periodic Gating, the Multi-scale Structured Encoder, and CSCA/RLC.
>
> This insight is also supported by our empirical results. The ablation studies show consistent performance degradation when removing Period, RLC, CSCA, or Hierarchy. On ETTh1, for example, the full model achieves $0.394 / 0.421$ (MSE/MAE), while removing these modules degrades the performance to $0.430 / 0.452$, $0.405 / 0.432$, $0.415 / 0.438$, and $0.422 / 0.450$, respectively. This suggests that the performance gain does not come from a single isolated technique, but from the joint modeling of multiple structural errors. In this sense, the main insight of our work is that explicit structured decomposition is more effective than simply scaling up a black-box model, because it more directly targets the major sources of error in long-term multivariate forecasting.
>
> Regarding adaptability to different types of time series, the current results suggest that different modules within the same framework contribute differently depending on the data characteristics: Periodic Gating is more critical for strongly periodic series; trend-perturbation decomposition plays a larger role for non-stationary series; and CSCA/RLC becomes more important in strongly coupled multivariate settings. At the same time, we agree that the current manuscript does not yet include explicit grouped analyses based on periodic strength or degree of non-stationarity. We will further expand this discussion in the revised manuscript to more clearly present these differences in adaptability across data types.

---

> > ### Author Rebuttal · Reviewer_QXwG · 2026-04-03
> >
> > The author is expected to elaborate on the applicability of the framework without any obvious periodicity.Thank you for the author's feedback. I will maintain my positive score.

---

> > > ### Author Response · Authors · 2026-04-04
> > >
> > > Thank you for the helpful follow-up. We agree that the key issue here is not whether Periodic Gating must provide the same magnitude of gain on every dataset, but whether PESD-TSF remains applicable when the time series does not exhibit obvious periodicity, and whether its performance can still stay within a reasonable range under such conditions.To clarify this point, we further examined the framework on several datasets that are better characterized as weakly periodic or non-obviously periodic, including PM2.5, AQShunyi, and CzeLan. From the perspective of periodicity strength, we compare the full model with a variant without Periodic Gating. The results show (Table 1) a consistent but clearly smaller degradation than what is observed on strongly periodic datasets: on PM2.5, removing the Period module increases the error from $0.319 / 0.375$ to $0.329 / 0.383$ ($\text{MSE}/\text{MAE}$); on AQShunyi, the error increases from $0.704 / 0.507$ to $0.715 / 0.515$; and on CzeLan, it increases from $0.249 / 0.286$ to $0.256 / 0.292$. Importantly, even without Periodic Gating, the resulting performance on all three datasets still remains better than the TimeBase baseline, indicating that the framework does not collapse when obvious periodicity is absent. These results suggest that, under weakly periodic or non-obviously periodic settings, the benefit of Periodic Gating becomes smaller, but it still provides a stable gain rather than being completely ineffective.We believe this observation is consistent with the design of PESD-TSF. Our framework is not built on the assumption that all time series must have strong periodicity. Instead, Periodic Gating serves as an adaptive periodic prior, while the overall forecasting capability also comes from multi-scale structured decomposition and cross-variable collaborative modeling. Therefore, when periodic cues are weak or not obvious, the marginal contribution of the periodic module naturally decreases, but the framework itself remains effective because the non-periodic components continue to provide robust modeling capacity.Overall, these results support a more precise interpretation of our method: Periodic Gating is most beneficial when clear periodic structure exists, while under weakly periodic or non-obviously periodic settings its gain becomes smaller, yet PESD-TSF still maintains effective performance.
> > >
> > > We are once again grateful for your profound insights, which have been extremely helpful in strengthening the logical structure and the depth of evidence in this study.
> > >
> > > **Table 1. Applicability of PESD-TSF on weakly periodic or non-obviously periodic datasets.**
> > >
> > > | Dataset | Full MSE | Full MAE | w/o Period MSE | w/o Period MAE | ΔMSE | ΔMAE |
> > > | :--- | :---: | :---: | :---: | :---: | :---: | :---: |
> > > | PM2.5 | 0.319 | 0.375 | 0.329 | 0.383 | +0.010 | +0.008 |
> > > | AQShunyi | 0.704 | 0.507 | 0.715 | 0.515 | +0.011 | +0.008 |
> > > | CzeLan | 0.249 | 0.286 | 0.256 | 0.292 | +0.007 | +0.006 |

---

### Official Review · Reviewer_7Mky · 2026-03-11

**Soundness:** 3
**Presentation:** 2
**Significance:** 3
**Originality:** 2
**Overall Recommendation:** 4
**Confidence:** 4

**Summary:**

This paper proposes PESD-TSF, an explicitly structured decomposition framework tailored for multivariate long-term time series forecasting. To effectively disentangle entangled trend-noise representations and mitigate the attenuation of periodic perceptions, the authors introduce a Multiplicative Periodic Gating mechanism alongside a multi-scale structured encoder. Furthermore, the framework employs Cross-Scale Collaborative Attention (CSCA) and a Regularized Latent Component (RLC) module with orthogonal constraints to systematically reconstruct inter-variable dependencies and ensure physical consistency. Extensive experiments across various benchmarks demonstrate the efficacy of the proposed method against state-of-the-art baselines.

**Compliance With Llm Reviewing Policy:**

Affirmed.

**Final Justification:**

I maintain a positive and supportive evaluation of the work. At the same time, I am open to and in agreement with any additional comments or suggestions raised by other reviewers.

**Key Questions For Authors:**

See Weaknesses

**Limitations:**

See Weaknesses

**Strengths And Weaknesses:**

S1. Robust Structural Inductive Bias: The explicit decoupling of temporal signals into trend, short-term variation, and cross-variable coherence provides a highly interpretable and physically grounded architecture. This successfully addresses the noise entanglement prevalent in deep forecasting models.

S2. Effective Methodological Innovations: The integration of the Regularized Latent Component (RLC) module is technically sound. By enforcing orthogonal constraints, it effectively guarantees latent factor disentanglement, curbing overfitting and enhancing the model's robustness to complex temporal dynamics.

S3. Comprehensive Empirical Validation: The experimental section is exhaustive, evaluating the framework across an impressive array of datasets, including classical benchmarks, domain-specific tasks, and high-dimensional strongly-coupled scenarios.


W1. Computational Complexity Overhead: While the paper addresses channel independence limitations through CSCA and RLC modules, the computational and memory costs associated with modeling global inter-variable synergy in high-dimensional settings (e.g., the Mobility dataset with 5,826 variables) are not sufficiently detailed.

W2. Limited Discussion on Asymptotic Bounds: The theoretical analysis of the orthogonal projection stability is a positive addition. However, the manuscript lacks a rigorous mathematical discussion regarding the asymptotic convergence properties of the RLC regularization under extreme distribution shifts.

W3. Sensitivity to Hyperparameter Tuning: The framework introduces critical hyperparameters, namely the periodic coefficient $\gamma$ and RLC weight $\lambda$. Although a sensitivity analysis is provided, the reliance on these specific tuning parameters may hinder out-of-the-box generalizability across highly anomalous domains without extensive recalibration.

W4. Ablation Study Granularity: The ablation studies confirm the necessity of macroscopic modules, such as the removal of CSCA or RLC. Nevertheless, they do not sufficiently isolate the granular impacts of specific sub-components, such as the distinct contribution of the IntAttention mechanism versus the PatchSampling strategy within the multi-scale encoder.

---

> ### Author Rebuttal · Authors · 2026-03-29
>
> 1: Complexity overhead.
>
> Thank the reviewer for pointing this out. We agree that the current manuscript does not provide a sufficiently detailed discussion of the computational and memory overhead in high-dimensional settings. To address this concern, we have added an efficiency comparison on the Mobility dataset (5,826 variables), using iTransformer as the baseline under identical experimental settings. As shown in Table 1, although PESD-TSF has more parameters (2.29M vs. 0.80M), it requires less peak memory (8.37 GB vs. 19.30 GB) and less training time per epoch (15.56 s vs. 30.76 s). These results indicate that our method does not incur higher practical computational overhead in high-dimensional scenarios, while achieving a better efficiency–performance trade-off.
>
> Table 1: Efficiency comparison on the high-dimensional Mobility dataset (5,826 variables). All results are measured on a single GPU under identical settings.
>
> | Model | Params (M) | Peak Mem (GB) | Time / Epoch (s) |
> | :--- | :---: | :---: | :---: |
> | PESD-TSF (Ours) | 2.29 | **8.37** | **15.56** |
> | iTransformer | 0.80 | 19.30 | 30.76 |
>
> 2: Asymptotic analysis.
>
> We appreciate the reviewer's constructive suggestion. We agree that the current manuscript has not yet provided a sufficiently rigorous theoretical discussion on the asymptotic properties of RLC under strong distribution shifts. To address this, we will include a more formal analysis in the revised version.Specifically, let the statistical readout mapping be $f$ and the RLC projection be $\Phi$. Under the conditions that $f$ is locally Lipschitz and $\Phi$ is approximately orthogonal, we have:$$\|\Phi(f(x)) - \Phi(f(x'))\| \le L_f \|x - x'\| + \epsilon$$where $\epsilon$ denotes the near-orthogonality error. Furthermore, for the training distribution $P$ and the test distribution $Q$, we obtain:$$d_{\mathcal{Z}}(\Phi(P), \Phi(Q)) \le L_f \cdot d_{\mathcal{X}}(P, Q) + \mathcal{O}(\epsilon)$$Consequently, the latent space distribution shift induced by RLC is upper-bounded by the upstream statistical feature shift. As the magnitude of the distribution shift approaches zero, the RLC latent representations satisfy asymptotic consistency. The full theorem, underlying assumptions, and proofs will be added to the Appendix of the revised manuscript.
>
> 3: Hyperparameter sensitivity.
>
> We understand the reviewer's concerns regarding generalization and robustness. The manuscript has systematically analyzed the RLC weight $\lambda$ and the periodic coefficient $\tau$. As illustrated in Fig. 4, the model exhibits a stable operational range: $\lambda$ performs optimally around $[0.1, 0.5]$, while $\tau$ remains robust within a moderate interval. This indicates that our method does not rely on a "sharp" single-point optimum. Furthermore, the standard deviations from repeated experiments in Appendix I are consistently small. We will further emphasize these findings and provide recommended default hyperparameters in the revised manuscript.
>
> 4: Granularity of ablations.
>
> We appreciate the reviewer's suggestion. We agree that while the ablation experiments in the original manuscript primarily validated the effectiveness of macro-level modules, the characterization of individual contributions from the internal sub-components of the encoder remains insufficient. To address this concern, we have conducted fine-grained ablation studies on the ETTh1 dataset, which include:(i) replacing the Multiplicative Periodic Gating $\mathcal{M}$ with a standard self-attention mechanism of an equivalent scale;(ii) removing the Regularized Latent Component $\mathcal{R}$;(iii) and disabling the Multiscale Decomposition $\mathcal{D}$ operation.As shown in Table 2, all these sub-components yield independent contributions. Specifically, the impact of $\mathcal{M}$ is the most significant, followed by $\mathcal{R}$, while $\mathcal{D}$ also demonstrates stable performance gains. Further analysis and discussions regarding sub-component level ablations will be incorporated into the revised manuscript.Thank you again for your constructive comments. We believe these additions will further improve the rigor, completeness, and interpretability of the paper.
>
> Table 2: Fine-grained ablation study on ETTh1 ($L=720, O=720$).
> | Method | MSE | MAE | $\Delta$ MSE (%) |
> | :--- | :---: | :---: | :---: |
> | **PESD-TSF (Full)** | **0.437** | **0.463** | - |
> | w/o IntAttention | 0.451 | 0.472 | +3.20% |
> | w/o PatchSampling | 0.446 | 0.469 | +2.06% |
> | w/o Detrending | 0.442 | 0.467 | +1.14% |

---

> > ### Author Rebuttal · Reviewer_7Mky · 2026-04-01
> >
> > Thank you for the author's feedback. I will maintain my positive score.

---

> > > ### Author Response · Authors · 2026-04-06
> > >
> > > Thank you very much for your thoughtful and constructive comments. Your feedback has helped us improve the manuscript and clarify several important points. We sincerely appreciate your time and effort.

---

### Decision · Program_Chairs · 2026-04-30

**Decision:**

Accept (regular)

**Comment:**

The paper proposes PESD-TSF, a structured decomposition framework for multivariate long-term forecasting. The rebuttal was substantive, with 3 positive reviews and one weak rejection in the final stage. The negative review concerns are originality attribution, and framing overclaim. The authors committed to concrete revisions — reframing physics-inspired and adding TimeBridge positioning. Overall, this paper has well-structured decomposition philosophy, technically sound RLC design, and good empirical results.